# Multi-level inhibition of coronavirus replication by chemical ER stress

Mohammed Samer Shaban[1,11], Christin Müller [2,11], Christin Mayr-Buro[1,11], Hendrik Weiser[1], Johanna Meier-Soelch[1], Benadict Vincent Albert[1], Axel Weber [1], Uwe Linne[3], Torsten Hain[4,5], Ilya Babayev[1], Nadja Karl[2], Nina Hofmann [6], Stephan Becker[7], Susanne Herold[8,9], M. Lienhard Schmitz [9,10], John Ziebuhr [2,5,12✉] & Michael Kracht [1,9,12✉]

Coronaviruses (CoVs) are important human pathogens for which no specific treatment is available. Here, we provide evidence that pharmacological reprogramming of ER stress pathways can be exploited to suppress CoV replication. The ER stress inducer thapsigargin efficiently inhibits coronavirus (HCoV-229E, MERS-CoV, SARS-CoV-2) replication in different cell types including primary differentiated human bronchial epithelial cells, (partially) reverses the virus-induced translational shut-down, improves viability of infected cells and counteracts the CoV-mediated downregulation of IRE1α and the ER chaperone BiP. Proteome-wide analyses revealed specific pathways, protein networks and components that likely mediate the thapsigargin-induced antiviral state, including essential (HERPUD1) or novel (UBA6 and ZNF622) factors of ER quality control, and ER-associated protein degradation complexes. Additionally, thapsigargin blocks the CoV-induced selective autophagic flux involving p62/SQSTM1. The data show that thapsigargin hits several central mechanisms required for CoV replication, suggesting that this compound (or derivatives thereof) may be developed into broad-spectrum anti-CoV drugs.

[1] Rudolf Buchheim Institute of Pharmacology, Justus Liebig University, Giessen, Germany. [2] Institute of Medical Virology, Justus Liebig University, Giessen, Germany. [3] Mass spectrometry facility of the Department of Chemistry, Philipps University, Marburg, Germany. [4] Institute of Medical Microbiology, Justus Liebig University, Giessen, Germany. [5] German Center for Infection Research (DZIF), partner site Giessen-Marburg-Langen, Giessen, Germany. [6] Bioinformatics and Systems Biology, Justus Liebig University, Giessen, Germany. [7] Institute of Virology, Philipps University, Marburg, Germany. [8] Department of Internal Medicine II for Pulmonary and Critical Care Medicine and Infectious Diseases, Justus Liebig University, and Institute for Lung Health (ILH), Giessen, Germany. [9] German Center for Lung Research (DZL) and Universities of Giessen and Marburg Lung Center (UGMLC), Giessen, Germany. [10] Institute of Biochemistry, Justus Liebig University, Giessen, Germany. [11] These authors contributed equally: Mohammed Samer Shaban, Christin Müller, Christin Mayr-Buro. [12] These authors jointly supervised this work: John Ziebuhr, Michael Kracht. ✉email: john.ziebuhr@viro.med.uni-giessen.de; michael.kracht@pharma.med.uni-giessen.de

Coronaviruses are enveloped plus-strand RNA viruses with a broad host range, including humans[1,2]. The four seasonal human CoVs (HCoV-229E, -NL63, -HKU1, -OC43) generally cause a spectrum of (mild) symptoms that are mainly restricted to the upper respiratory tract[3–6]. In contrast, the three highly pathogenic CoVs that emerged from animal reservoirs over the past two decades are frequently associated with significant disease burden and mortality in humans. The latter include the severe acute respiratory syndrome CoV (SARS-CoV)[7–9], SARS-CoV-2[10,11], and Middle-East respiratory syndrome CoV (MERS-CoV)[12].

The current SARS-CoV-2 pandemic highlights the urgent need to identify new antiviral strategies, including drugs that target the host side[10]. CoVs impose multiple functional but also structural changes to a wide range of cellular pathways and there is increasing evidence that some of these pathways may be exploited therapeutically[13,14].

In common with other cellular stress conditions, including infections by diverse pathogens, CoVs are known to activate the NF-κB, JNK, and p38 MAPK pathways and to reprogram host cell transcriptomes[15–17]. In addition, they induce the formation of replicative organelles (ROs), an intracellular network of convoluted membranes (CMs), and double-membrane vesicles (DMV) that harbor the viral replication/transcription complexes (RTC) and shield these complexes from recognition by cellular defense mechanisms[18]. The origins of the membranes used for RO are still under debate but can be linked, at least partly, to ER- and autophagy-related processes[19]. The combination of these and other events leads to cell damage and cell death upon virus budding and release within a few days[14]. The virus-induced cellular changes are associated with an activation of the unfolded protein response (UPR), which is evident from a profound transcriptomic endoplasmic reticulum (ER) stress signature, as reported for cells infected with HCoV-229E[17].

The ER is critically involved in surveying the quality and fidelity of membrane and secreted protein synthesis, as well as the folding, assembly, transport, and degradation of these proteins[20]. The accumulation of unfolded or misfolded proteins in the ER lumen leads to ER stress and UPR activation, thereby slowing down protein synthesis and increasing the folding capacity of the ER[21]. As a result, cellular protein homeostasis can be restored and the cell survives. If this compensatory mechanism fails, ER stress pathways can also switch their functions, inducing oxidative stress and resulting in cell death[20,22].

The system relies on sensors residing in the ER membrane which include the protein kinase R (PKR)-like ER kinase (PERK), inositol-requiring protein 1α (IRE1α), and cyclic AMP-dependent transcription factor 6α (ATF6α). PERK and IRE1α are Ser/Thr kinases whose conserved N termini are oriented towards the ER lumen[23]. In non-stressed cells, the highly abundant major ER chaperone and ER stress sensor binding-immunoglobulin protein BiP (also called 78 kDa glucose-regulated protein, GPR78; heat shock protein family A member 5, HSPA5) binds to PERK and IRE1α, which keeps these two proteins in an inactive monomeric state[24,25]. Upon increased binding of BiP to misfolded ER clients, BiP is released from both PERK and IRE1α, resulting in an (indirect) activation of the two kinases by oligomerization and trans(auto)phosphorylation[26–28].

Active PERK phosphorylates the eukaryotic translation initiation factor 2 (eIF2) subunit α to shut down translation and also activates ATF4, the master transcription factor orchestrating ER stress-induced genes[29,30]. Phosphorylated IRE1α activates its own RNase domain to generate spliced (s)XBP1 protein, a multifunctional transcriptional regulator responsible for adaptive responses or cell death[31]. The specific function(s) of yet another ER stress-activated transcription factor, ATF3, is less well understood[32]. Generally, the various branches of the UPR act in concert, allowing a multitude of potential outcomes, ranging from the compensation of ER stress and restoration of proteostasis to cell death[22].

The activation of ER stress by microbial and viral infections has been widely observed. However, with few exceptions, it remains to be studied how this response is shaped in an infectious agent-specific manner and whether or not these responses are beneficial or detrimental to the host[33]. Moreover, there is a lack of knowledge on CoV-mediated (de)regulation of ER stress components at the protein level. The latter is important because CoVs, in common with many RNA viruses, are known to cause a global shutdown of host protein synthesis[34].

Here, we report that CoV infection activates ER stress signaling and induces UPR components at the mRNA level while suppressing them at the protein level. Strikingly, the well-known chemical activator of the UPR, thapsigargin, exerts a profound antiviral effect in the lower nanomolar range on three different CoVs in four different cell types, including human primary bronchial epithelial cells. A detailed proteomics analysis revealed multiple thapsigargin-regulated pathways and a network of proteins involved in ER quality control (ERQC) or ER-associated degradation (ERAD) that are suppressed by CoV but (re)activated by chemically stressed infected cells. Additionally, we discovered that thapsigargin blocks the autophagic flux in CoV-infected cells. Taken together, these data provide important new insight into central factors involved in CoV replication and open new avenues for targeted CoV antivirals.

## Results

To investigate how CoVs modulate ER stress components at the mRNA compared to the protein level, we determined the expression levels of 166 components of the ER stress pathway KEGG hsa04141 "protein processing in ER" in human HuH7 liver cells, a commonly used cellular model for CoV replication, in response to infections with HCoV-229E and MERS-CoV, respectively. For untreated HuH7 cells, we obtained mRNA (by RNA-seq) and protein (by LC-MS/MS) expression data for 119 components which revealed a positive correlation between mRNA and protein abundances (Fig. 1a, upper graph). However, in cell lysates obtained at 24 h post infection (p.i.), this effect was largely lost (Fig. 1a, middle and lower graph). Pearson correlation matrix confirmed a progressive loss of correlation between mRNA levels and protein levels for this pathway over a time course from 3 to 24 h p.i. (Fig. 1b). Thus, out of 39 (for HCoV-229E) or 56 (for MERS-CoV) ER stress factors that were found to be regulated at the mRNA level, only a few remained regulated at the protein level at late time points (Fig. 1c, shown as a projection on KEGG hsa04141 in Supplementary Fig. 1).

To determine the functional consequences of this opposing regulation at the mRNA and protein levels in CoV-infected cells, we focused on HCoV-229E and assessed key regulatory features of the ER stress pathway as shown in Fig. 2a. As a reference, we included samples from cells exposed to thapsigargin, a compound that has been widely used to study prototypically activated ER stress mechanistically[25,35]. This setup included experiments, in which thapsigargin and virus were added simultaneously to the cell culture medium (followed by a further incubation for 24 h) or thapsigargin was added to the cells at 8 h p.i. for 16 h (Fig. 2b).

The presence of thapsigargin in the growth medium resulted in a major drop in viral titers by more than 150-fold (from $9.18 \times 10^6$ to $5.7 \times 10^4$ pfu/ml), which was paralleled by reduced amounts of viral RNA isolated from thapsigargin-treated, HCoV-229E-infected cells at 24 h p.i. (Fig. 2c). Immunofluorescence analysis of HCoV-229E-infected cells treated with thapsigargin

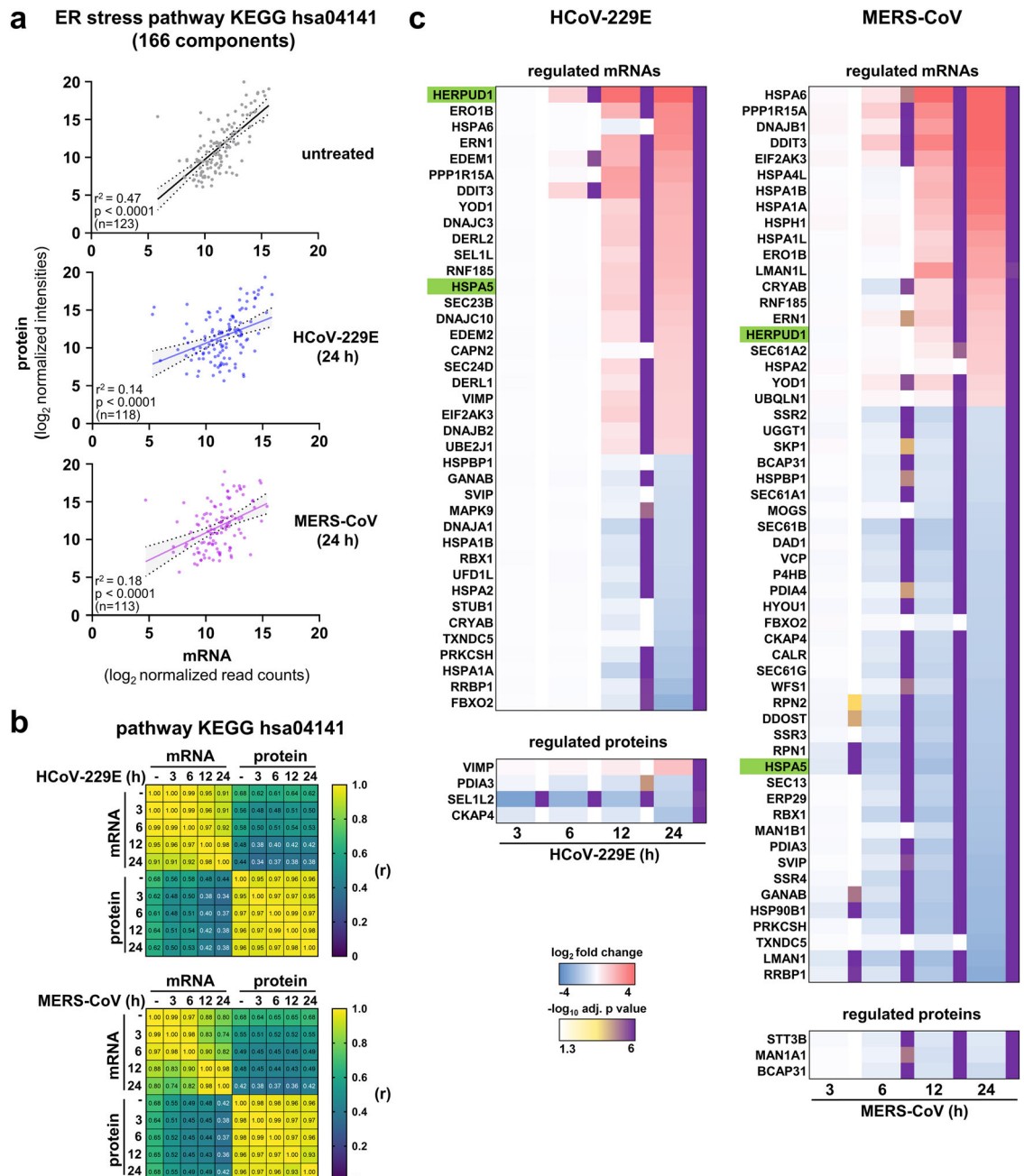

**Fig. 1 CoVs uncouple mRNA and protein levels of ER stress components in infected cells. a–c** HuH7 cells were left untreated or were infected with HCoV-229E or MERS-CoV (MOI = 1) for 3, 6, 12, or 24 h. Transcriptomic and proteomic data were derived from samples obtained at the indicated time points post infection (p.i.). Subsequently, mRNA and protein expression values for the KEGG pathway hsa04141 "protein processing in endoplasmic reticulum" from two biologically independent experiments were extracted and used for further analysis. **a** Scatter plots show mean normalized protein/ mRNA expression values for the number (*n*) of expressed components, fitted linear regression lines, 95% confidence intervals, and coefficients of determination for non-infected HuH7 cells and HuH7 cells infected with virus for 24 h. *P* values were calculated from an F test to test the null hypothesis that the overall slope is zero. **b** Correlation matrix of Pearson's r across all conditions. All corresponding p values are provided in the Source data. **c** The heatmaps show mean ratio values of differentially expressed mRNAs or proteins based on significant differences (fold change ≥ 2, adjusted $p \leq 0.01$) calculated from the replicates by moderated t-tests. See also Supplementary Fig. 1 for pathway mappings of mRNA and protein data and Source data for complete data sets.

confirmed the impaired formation of functional viral RTCs as shown by the reduced levels of both double-stranded RNA (an intermediate of viral RNA replication) and nonstructural protein (nsp) 8 (an essential part of the viral RTC) (Fig. 2d).

A strong suppression of viral replication was also demonstrated by the reduced protein levels observed for the nucleocapsid (N) protein (a major coronavirus structural protein) as well as nsp 8

and 12, both representing essential components of the viral replication complex[36] (Fig. 2e, f). In all cases, the antiviral effect of thapsigargin remained readily detectable when the compound was added at 8 h p.i., suggesting that it does not prevent viral entry but rather suppresses intracellular pathways required for efficient viral RNA replication and virus formation and release, or, activates unknown antiviral effector systems (Fig. 2c, e, f).

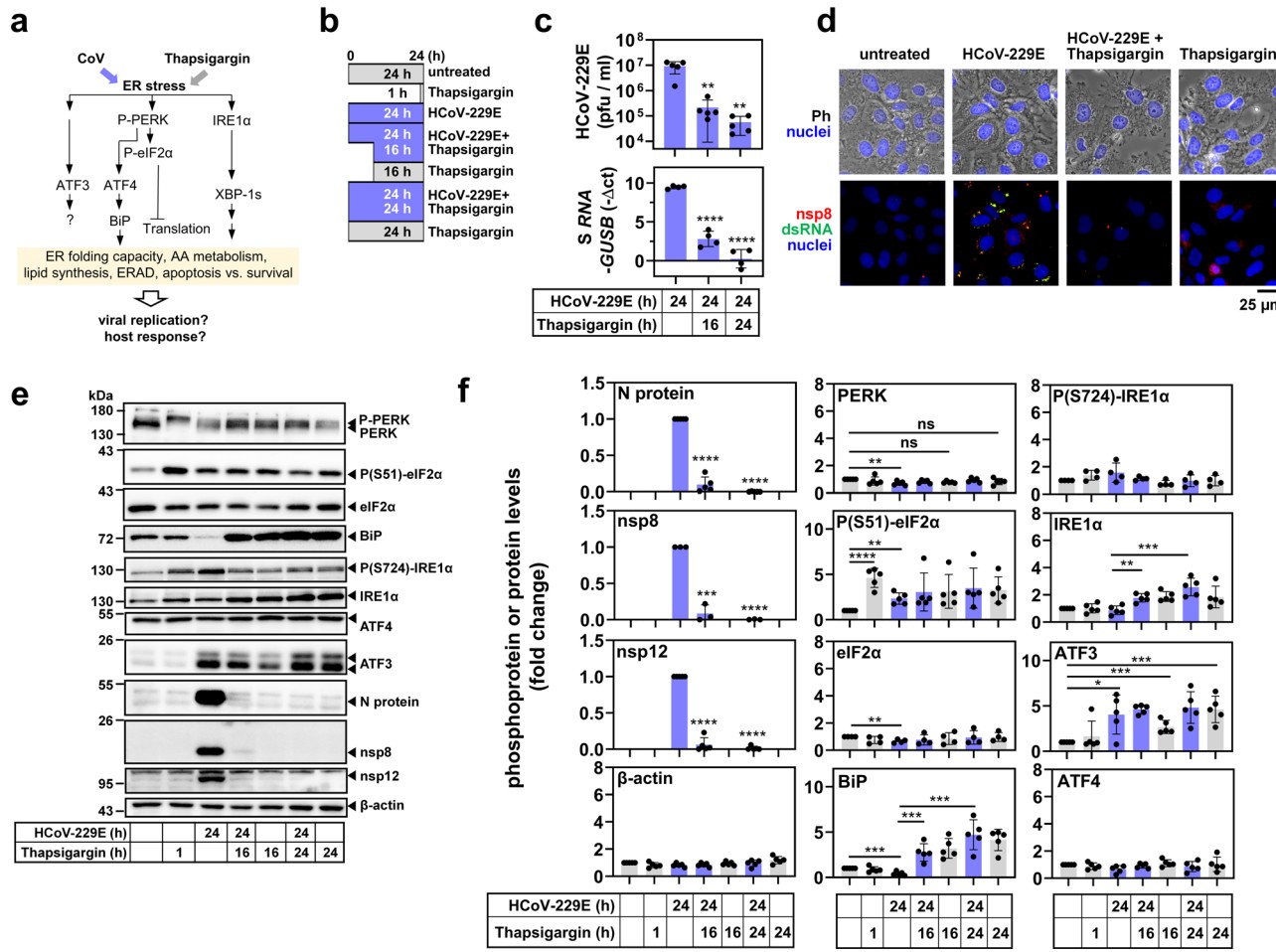

**Fig. 2 Thapsigargin inhibits HCoV-229E replication and counteracts virus-mediated BiP downregulation. a** Schematic overview of parameters used to monitor virus- and thapsigargin-mediated ER stress. **b** Schematic presentation of HCoV-229E infection of cells and/or treatment with thapsigargin as applied in this study. **c** HuH7 cells were left untreated or infected with HCoV-229E (MOI = 1) for 24 h and treated with thapsigargin (1 μM) according to the scheme shown in (**b**). Supernatants and RNA isolated from the cell pellets were used to determine viral titers by virus plaque assay and expression of HCoV-229E S gene-encoding RNA (five (upper graphs) or four (lower graphs) biologically independent experiments). **d** Phase-contrast (Ph) and fluorescence microscopy images showing the morphology of HuH7 cells and the subcellular HCoV-229E replication sites (at 24 h p.i.) identified by nsp8- and double-strand RNA-specific antibodies in the presence or absence of thapsigargin (1 μM for 24 h) (representative images from one out of two biologically independent experiments). Nuclei were stained with Hoechst 33342. **e** Representative immunoblots of total cell extracts from HuH7 cells infected with HCoV-229E (MOI = 1) and treated with thapsigargin (1 μM) according to (**b**) showing the expression/modification of the indicated host cell and viral proteins. **f** Quantification of immunoblot data as shown in (**e**) relative to the untreated control (three or more biologically independent experiments). All bar graphs show means ± s.d.; asterisks indicate $p$ values (*$p \leq 0.05$, **$p \leq 0.01$, ***$p \leq 0.001$, ****$p \leq 0.0001$) obtained by two-tailed unpaired t-tests.

Next, we investigated ER stress signaling under these conditions. Both virus and thapsigargin were confirmed to activate the PERK branch of ER stress (Fig. 2e, f), as shown by the retarded mobility of PERK in SDS gels (indicating multisite phosphorylation) and by phosphorylation of the PERK substrate eIF2α at Ser51 (Fig. 2e, f). HCoV-229E infection led to a weak but significant decrease of PERK (mean $71 \pm 15\%$) and eIF2α (mean $67 \pm 13\%$) levels compared to the controls. Infection also caused an approximately twofold (mean $42 \pm 22\%$) reduction of BiP expression (Fig. 2e, f). In contrast, long-term thapsigargin treatment (for 16 h or 24 h) caused a 3–4-fold increase of BiP expression, also in HCoV-229E-infected cells, thus reversing the suppression by viral infection (Fig. 2e, f). Similarly, thapsigargin treatment for 16 h or 24 h caused a 1.5–2-fold increase in IRE1α expression (but not phosphorylation), again also in infected cells (Fig. 2e, f). In this set of experiments, ATF3 proved to be the only protein that was induced by the virus alone (Fig. 2e, f), while the expression levels of ATF4 remained largely unchanged (Fig. 2e, f).

To reveal the role of PERK in these effects, we treated cells with the protein kinase inhibitor GSK2656157. This compound suppressed PERK autophosphorylation, PERK activity (on eIF2α), and CoV replication without having a major impact on cell viability (Supplementary Fig. 2a–c). However, inhibition of viral replication by GSK2656157 was less effective than thapsigargin and, in combined treatments, it did not abolish the thapsigargin-mediated suppression of N protein expression and virus replication, placing the thapsigargin-mediated viral suppression downstream of PERK (Supplementary Fig. 2d, e).

These data show that both CoV infection and chemicals like thapsigargin activate ER stress through the same proximal PERK pathway, although they affect downstream cellular outcomes differentially. The restoration of BiP and IRE1α levels by long-term thapsigargin treatment further suggests that the CoV-induced block of inducible host factors is reversible and can be reprogrammed by a (presumably protective) thapsigargin-mediated response. Our comparative analyses of viral replication and host

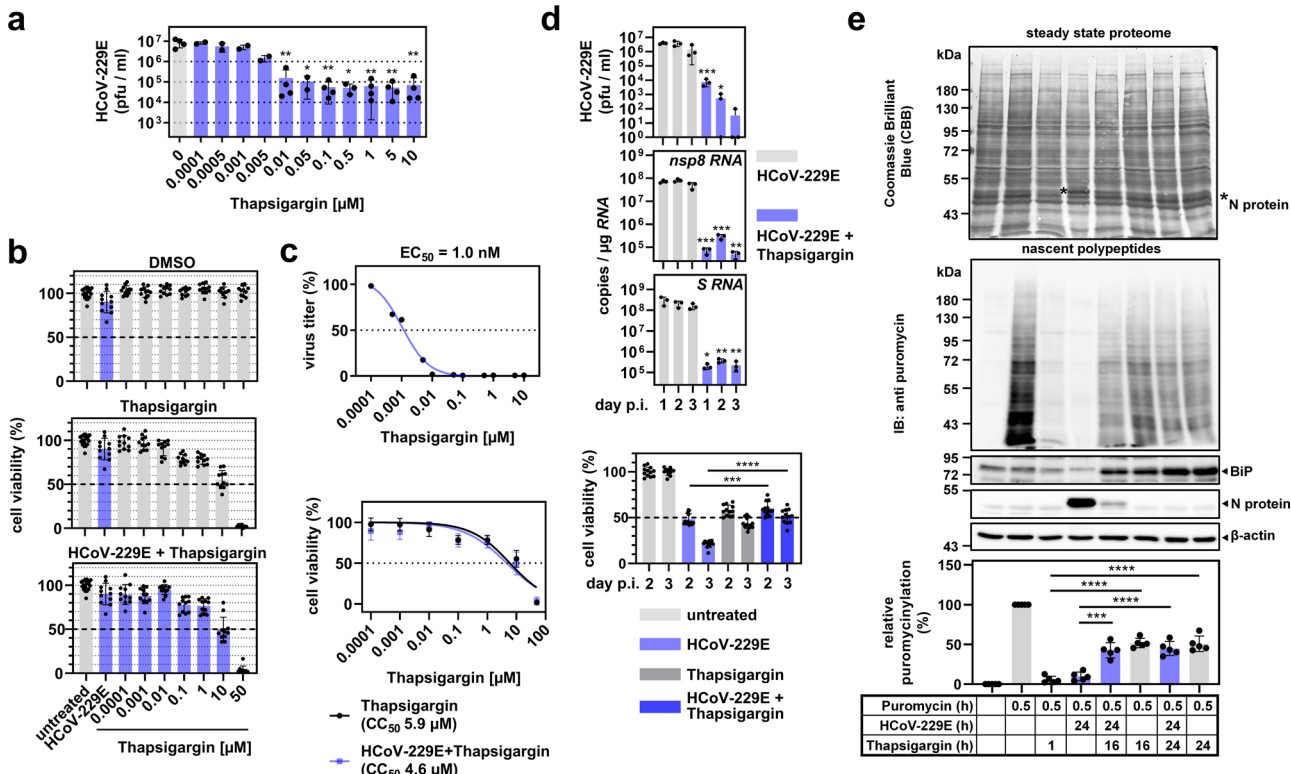

**Fig. 3 Thapsigargin mediates dose-dependent, long-lasting inhibition of HCoV-229E replication and leads to partially improved survival and protein biosynthesis of infected cells. a** Dose-dependent suppression of HCoV-229E replication in the absence or presence of increasing concentrations of thapsigargin. At 24 h p.i., viral titers of cell culture supernatants were determined (two or more biologically independent experiments). **b** HuH7 cells were left untreated, or were infected with HCoV-229E (MOI = 1) in the presence/absence of increasing concentrations of thapsigargin or DMSO as solvent control. After 24 h, cell viability was assessed by MTS assay (five biologically independent experiments). **c** All data from (**a**) and (**b**) were used to compute the relative estimated half-maximal effective ($EC_{50}$) (upper graph, means of two or more biologically independent experiments) and cytotoxic ($CC_{50}$) (lower graph, means ± s.d. of five biologically independent experiments) concentrations of thapsigargin in HuH7 cells infected with HCoV-229E. **d** HuH7 cells were infected with HCoV-229E (MOI = 1) and treated with a single dose of thapsigargin (1 μM) as indicated. Viral titers, copy numbers of viral RNAs (detected using *nsp8* coding sequence- and *S* gene-specific primers, respectively), and cell viabilities were determined after 1–3 days (three biologically independent experiments). **e** HuH7 cells were treated and infected according to Fig. 2b. Thirty minutes before harvesting, puromycin (3 μM) was added where indicated. Total cell extracts were analyzed by immunoblotting. The blot membrane was stained with CBB to assess the steady-state proteomes and then hybridized with anti-puromycin antibodies to detect de novo synthesized nascent polypeptides. Puromycin signals of each lane were normalized to the corresponding CBB staining and were background corrected by subtracting signals of samples in which puromycin had been omitted. The upper graphs show representative images and the lower graph shows the quantification of five biologically independent experiments. All bar graphs show means ± s.d.; asterisks indicate *p* values (*$p \leq 0.05$, **$p \leq 0.01$, ***$p \leq 0.001$, ****$p \leq 0.0001$) obtained by two-tailed unpaired t-tests. See Supplementary Fig. 2 for PERK inhibitor data.

response, along with the effects of PERK inhibition, lead us to conclude that chemically and virus-induced forms of ER stress, although proceeding through the same core PERK pathway, do not simply potentiate each other but rather (somewhat counter-intuitively) counteract each other.

To explore a potential pharmacological exploitation of this effect, we assessed the (half-maximal) effective and cytotoxic concentrations ($EC_{50}$, $CC_{50}$) of the combined thapsigargin treatment and virus infection, because both conditions are known to promote cell death. HCoV-229E replication was suppressed with an $EC_{50}$ of 1 nM (Fig. 3a, c), as judged by virus titration of cell culture supernatants obtained from these cells. At 24 h p.i., the cell viability of HCoV-229E-infected HuH7 cells was only marginally reduced (mean 90.02 ± 12.32%) (Fig. 3b, upper graph). After 24 h of incubation, thapsigargin decreased cell viability in a dose-dependent manner with a $CC_{50}$ of 5.9 μM in line with previous reports (Fig. 3b, middle graph, Fig. 3c)[37,38]. The combination of thapsigargin treatment and HCoV-229E infection did not cause additional cytotoxicity as shown by a nearly identical $CC_{50}$ of 4.6 μM (Fig. 3b, lower graph, Fig. 3c). At 1 μM

thapsigargin, i.e., a concentration shown to completely abolish viral protein translation and production of infectious virus progeny (see above), the cell viability of cells infected with HCoV-229E and treated with thapsigargin was 76.6 ± 7.9% (Fig. 3b, c). Furthermore, the antiviral effects of thapsigargin remained detectable for three days after a single dose, with a profound reduction of viral titers and RNA levels by several orders of magnitude (Fig. 3d). After three days, 50% of the cells infected with HCoV-229E and treated with thapsigargin were still viable, compared to only 20% of cells that survived the infection in the absence of thapsigargin (Fig. 3d). The data suggest that thapsigargin exerts long-lasting antiviral effects at concentrations well below its cytotoxic concentrations.

To further characterize the metabolic state of the cells under the conditions used in these experiments, we investigated protein de novo synthesis. Newly produced proteins were quantified by in vivo puromycinylation tagging of nascent protein chains followed by immunoblotting using anti-puromycin antibodies. HCoV-229E infection was found to shut down cellular protein biosynthesis by 90.3 ± 5.4%, while treatment with thapsigargin for

1 h led to a translational shut-down by $94.3 \pm 4.3\%$ (Fig. 3e). However, in infected cells, the simultaneous or delayed addition of thapsigargin restored (or rescued) protein biosynthesis to approximately 50% of the level observed in untreated cells (Fig. 3e). These data demonstrate that, although both viral infection and thapsigargin treatment (individually) induce ER stress and cause a translational shut-down, their combination shows no additive harmful effects on the cells. On the contrary, their combination appears to have opposing effects that result in a partial restoration of the cellular metabolic capacity while retaining a profound antiviral effect.

We next assessed if these effects were cell type or virus-specific. In line with the results described above, the antiviral effects of thapsigargin, the reconstitution of the BiP and IRE1α levels, and the lack of additional cytotoxicity in infected cells could be confirmed for diploid MRC-5 embryonic lung fibroblasts infected with HCoV-229E (Fig. 4a–d), as well as for HuH7 cells infected with MERS-CoV and Vero E6 African green monkey kidney epithelial cells infected with SARS-CoV-2 (Fig. 4e–k and Supplementary Fig. 3a, b). MERS-CoV and SARS-CoV-2 replication were suppressed by thapsigargin with an $EC_{50}$ of 4.8 and 260 nM, respectively (Fig. 4i, j), while the $CC_{50}$ for thapsigargin in Vero E6 cells was 18.25 μM (based on MTT assay) or 20.27 μM (based on ATPlite assay) (Fig. 4k), resulting in selectivity indices (SI, $CC_{50}/EC_{50}$) of 1229 for MERS-CoV and 70 (MTT) to 78 (ATPlite) for SARS-CoV-2, respectively. In contrast to thapsigargin, the inhibition of PERK by GSK2656157 required higher concentrations in the low micromolar range to attain a significant drop of MERS-CoV replication, once again supporting the exceptional efficacy of thapsigargin (Supplementary Fig. 2f).

Next, we tested potential antiviral activities of thapsigargin against other RNA viruses and used (as a reference) remdesivir, an adenosine analog that inhibits coronavirus RNA-dependent RNA polymerases[39]. We found that thapsigargin suppressed influenza A virus (IAV) but not poliovirus replication (Supplementary Fig. 4a, b). As shown in Supplementary Fig. 5a–c, remdesivir reduced the replication of HCoV-229E ($EC_{50} < 10$ nM), MERS-CoV ($EC_{50} = 5.3$ nM), and SARS-CoV-2 ($EC_{50} = 2.38$ μM) in HuH7 and Vero E6 cells at non-toxic concentrations (Supplementary Fig. 5d, e), in line with previously published studies[40–42]. The data show that thapsigargin has potent antiviral activity against another family of enveloped RNA viruses and is at least as effective as remdesivir against HCoV-229E and MERS-CoV, while thapsigargin is approximately 10-times more effective than remdesivir against SARS-CoV-2.

To corroborate these observations in a physiologically more relevant system, we established cell cultures of differentiated normal human bronchial epithelial (NHBE) cells[43]. After the initial expansion, the cells were exposed to air-liquid interfaces allowing their differentiation into various airway cell types, which was validated by fluorescence microscopy using antibodies detecting tight junctions and marker proteins specific for the goblet, ciliated and basal cells, respectively (Fig. 5a, b). As shown in Fig. 5c, thapsigargin inhibited the replication of all three coronaviruses included in this experiment (HCoV-229E, MERS-CoV, SARS-CoV-2) in a dose-dependent manner in NHBE cells obtained from different donors. Cell viability following thapsigargin treatment (as judged by measuring the integrity of the epithelial monolayer using TEER) ranged between 70 and 80% after 72 h (Supplementary Fig. 5f). Similar to the cell lines used before, SARS-CoV-2 was found to be slightly less sensitive to thapsigargin treatment. Importantly, in the presence of 1 μM thapsigargin, no infectious virus progeny of any of the three CoVs was detectable at later time points p.i. (Fig. 5c).

To characterize the underlying molecular mechanisms responsible for the observed antiviral effects of thapsigargin, we focused on the two highly pathogenic coronaviruses, MERS-CoV

and SARS-CoV-2, for which, to our knowledge, no side-by-side comparison of proteomic changes has been reported at the time of our study. The large-scale proteomic study included (i) untreated cells and cells that were (ii) infected with MERS-CoV, (iii) infected with SARS-CoV-2, (iv) treated with thapsigargin, or, (v and vi) infected with one of these viruses in the presence of thapsigargin. We used label-free quantification to determine the expression levels of >5000 protein IDs from total cell extracts.

In a systematic approach, we identified differentially expressed proteins (DEPs) based on pairwise comparisons of proteins obtained from untreated cells, virus-infected cells, or thapsigargin-treated cells using a $p$ value of $-\log_{10}(p) \geq 1.3$ as cut-off. As visualized by Volcano plot representations, MERS-CoV infection suppressed 412 (at 12 h p.i.) and 1171 proteins (at 24 h p.i.), respectively, and increased the levels of 150 proteins (at 12 h p.i.) and 508 proteins (at 24 h p.i.), respectively (Fig. 6a, b), while SARS-CoV-2 suppressed the expression of 250 proteins at 12 h p.i. and 159 proteins at 24 h p.i. and increased the expression of 224 proteins at 12 h p.i. and 63 proteins at 24 h p.i. (Fig. 6c, d). Thapsigargin treatment alone suppressed/induced large numbers of proteins in HuH7 cells (918 down, 893 up at 12 h; 1711 down, 958 up at 24 h) and in Vero E6 cells (225 down, 191 up at 12 h; 249 down, 162 up at 24 h) (Fig. 6a–d). As expected, this analysis also identified viral proteins as the most strongly regulated DEPs. A comparison of virus-infected cells with virus-infected cells treated with thapsigargin revealed a complete suppression of all viral proteins and a large number of proteins with increased expression in thapsigargin-treated cells infected with MERS-CoV (843, 12 h p.i.; 1208, 24 h p.i.; red groups of proteins) or SARS-CoV-2 (299, 12 h; 362, 24 h; red groups of proteins) (Fig. 6a–d, right graphs). Also, similar numbers of proteins were identified with higher expression in virus-infected cells compared to virus-infected cells treated with thapsigargin (Fig. 6a–d, right graphs; blue groups of proteins). Together, these data lead us to conclude that thapsigargin causes a profound shift in protein expression in infected cells that likely contributes to the antiviral effects of this compound.

We then devised a bioinformatics strategy to identify patterns of co-regulated or unique pathways and link deregulated protein sets identified in these data to specific (known) biological functions. As shown schematically in Fig. 6e, we sorted the DEPs from each of the four groups shown in Fig. 6a–d into four multiple gene ID lists, annotated the gene IDs to biological pathways, and generated hierarchically clustered heatmaps of the top 100 differentially enriched pathway categories for the 12 h p.i. and 24 h p.i. time points of MERS-CoV- and SARS-CoV-2-infected cells, respectively, versus thapsigargin-treated cells by over-representation analysis (ORA) using Metascape software[44]. In this analysis, the groups of up- or downregulated proteins were kept separate to preserve information on whether specific DEPs belonging to particular overrepresented pathway terms were regulated in the same or opposite direction. Inspection of the four top 100 clustered heatmaps shows many similarities but also differences in pathways and their enrichment $p$ values in response to virus infection or thapsigargin, with the combined data revealing the complexity of cellular responses to CoV infections or chemical stressors (Supplementary Figs. 6, 7). By condensing this information to the top 5 pathways for up- or downregulated DEPs, we found that many of the most highly enriched categories are related to RNA, DNA, metabolic functions and localization (Supplementary Fig. 8a). We then combined the 400 pathway categories and searched this list for identical or unique GO terms in response to MERS-CoV, SARS-CoV-2, or thapsigargin. By filtering 229 pathways (out of 400) with enrichment $p$ values of $\log_{10}(p) \leq -3$, we found 36 pathway categories shared by both viruses and by thapsigargin, which are mostly related to RNA,

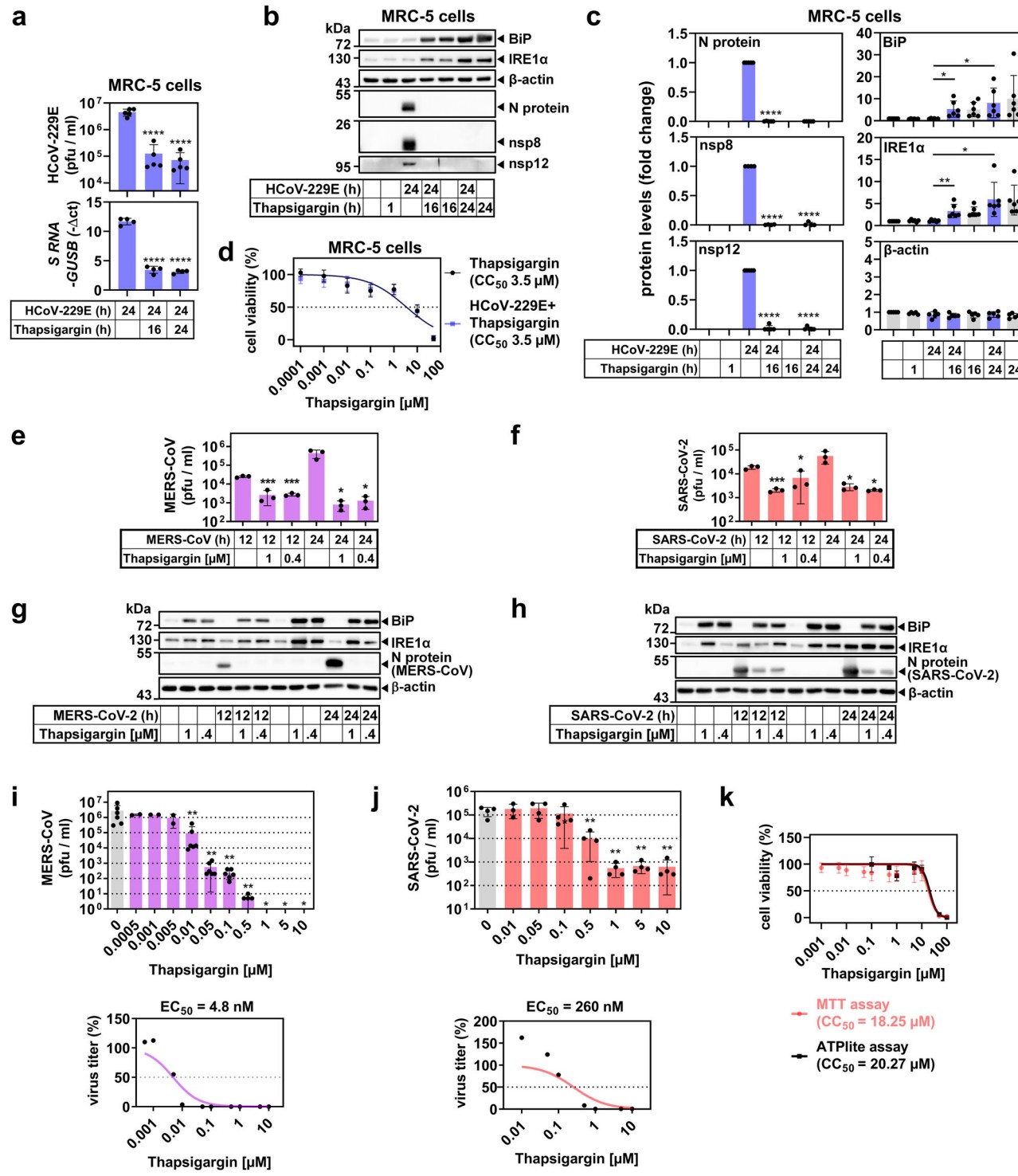

folding, stress, and localization (Fig. 6f and Supplementary Fig. 8b). Fifty-two pathway categories unique to thapsigargin almost exclusively represented metabolic and biosynthetic pathways as shown for the top 20 overrepresented pathways containing up- or downregulated DEPs, suggesting that thapsigargin on its own, unlike CoV infection, initiates a broad metabolic response (Fig. 6f and Supplementary Fig. 8b).

This raised the question of whether the thapsigargin effects were retained in infected cells or, alternatively, drug-sensitive pathway patterns were reprogrammed (or masked) by the virus infection. To address this point, we pooled all pathways enriched

under virus + thapsigargin conditions and compared them to virus infection or thapsigargin treatment alone. 59% (147 out of 249) pathway terms were shared by these three conditions, reflecting multiple stress-related catabolic, RNA regulatory, and vesicle or autophagy processes (Fig. 6g, h). 20 pathway terms were unique to the virus + thapsigargin situation. They primarily mapped to specific splicing, signaling (TORC, RHOA, ARF3) and transport/localization pathways (Fig. 6g, h). The 37 categories shared by virus + thapsigargin and thapsigargin conditions but not detectable in cells infected with virus (only) recapitulate the thapsigargin-regulated metabolic pathways (metabolism of

**Fig. 4 Thapsigargin inhibits the replication of high- and low-pathogenic human coronaviruses in multiple cell types. a–d** Human embryonic MRC-5 lung fibroblasts were infected with HCoV-229E according to the scheme shown in Fig. 2b. **a** Viral titers (upper graph, five biologically independent experiments) and expression of viral *S* gene-encoding RNAs (lower graph, four biologically independent experiments). **b, c** Expression of viral and host cell proteins. **b** Shows one representative immunoblot of total cell extracts and **c** shows quantification from four or more biologically independent experiments. **d** Cell viability was analyzed and quantified as described in the legend of Fig. 3 (four biologically independent experiments). **e–j** Similarly, HuH7 cells or Vero E6 African green monkey kidney epithelial cells were infected with MERS-CoV (MOI = 0.5) or SARS-CoV-2 (MOI = 0.5) for 12 h or 24 h in the presence/ absence of 0.4 μM or 1 μM thapsigargin. **e, f** Show viral titers and **g, h** display representative images of the corresponding expression of MERS-CoV/SARS-CoV-2 nucleocapsid (N) and host cell proteins, respectively (three biologically independent experiments). **i** Dose-dependent suppression of MERS-CoV-2 replication by thapsigargin in HuH7 cells infected with an MOI of 0.5 (upper graph, two or more biologically independent experiments) and the estimated $EC_{50}$ concentration calculated from the mean values (lower graph). **j** Dose-dependent suppression of SARS-CoV-2 replication by thapsigargin in Vero E6 cells infected with an MOI of 0.5 (upper graph, three or more biologically independent experiments) and the estimated $EC_{50}$ concentration from the mean values (lower graph). **k** The $CC_{50}$ of thapsigargin in Vero E6 cells was calculated by MTT or ATPlite assays as described in the legend of Fig. 3b, c. Data are from three or more biologically independent experiments. All bar graphs show means ± s.d.; asterisks indicate *p* values (*$p \leq 0.05$, **$p \leq 0.01$, ***$p \leq 0.001$, ****$p \leq 0.0001$) obtained by two-tailed unpaired t-tests (**a, c, e, f, j**) or ordinary one-way ANOVA (**i**). See Supplementary Figs. 3–5 for quantifications from replicates for MERS-CoV/SARS-CoV-2 immunoblot experiments and for further inhibitor data.

monosaccharide, cysteine, glutathione, methionine, fructose, mannose, pyruvate, TCA cycle, ERAD pathway) (Fig. 6g, h). For some of these pathways (e.g., ERAD, monocarboxylic acid metabolism), some DEPs were induced while others were repressed, indicating remodeling of pathway functions at the protein level (Fig. 6g, h). The 36 pathway terms that were absent in the virus + thapsigargin group of terms (groups 15, 9, 12 of the Venn diagram shown in Fig. 6g) represent a distinct set of terms, most of which being related to nucleotide and DNA-related processes, such as DNA repair, DNA unwinding, chromatin silencing (Fig. 6g, h). In summary, the functional analysis of DEPs at the level of differentially enriched pathway categories shows that the antiviral effects of thapsigargin strongly correlate with the activation/suppression of a range of metabolic programs.

The enriched pathway terms provided important overarching information on shared and unique biological processes but not necessarily encompassed identical sets of DEPs as exemplified by the twelve pathways shown in Fig. 6i. We, therefore, refined our analysis to the individual component level to identify proteins with similar regulation between both viruses across both cell types. The proteomes of HuH7 and Vero E6 cells overlap by 57% (Fig. 7a). In this group, only 43 identical proteins were found to be deregulated by both MERS-CoV and SARS-CoV-2 (Fig. 7b, left Venn diagrams). However, under thapsigargin + virus conditions, 120 proteins were upregulated and 63 proteins were downregulated (Fig. 7b, right Venn diagrams). Using the top 50 DEPs as an example, it becomes apparent that the majority of proteins are regulated into the same direction by thapsigargin alone; demonstrating that thapsigargin largely overrides any virus-induced modulation of host processes (Fig. 7c).

In the absence of thapsigargin, the virus infection generally has little or opposite effects on the levels of the 120 proteins, as exemplified by the suppression observed for BiP (HSPA5) or HERPUD1 (Fig. 7c, highlighted in green). The 120 induced factors map to pathways involving Golgi vesicle transport, ER stress, fiber organization, and apoptosis (Fig. 7d). Across their pathway annotations, 71 out of the 120 proteins were reported to strongly interact, thus probably being involved in protein:protein networks that coordinate activities of the enriched pathways (Fig. 7e, left graph). Likewise, the 63 repressed proteins map to specific (though different) pathways, such as monocarboxylic acid metabolism or viral life cycle (Fig. 7d). 26 components can be allocated to a few small protein interaction networks (Fig. 7e, right graph).

In our proteomic/bioinformatics analysis, HERPUD1 and p62/SQSTM1 were revealed to be among the most prominent thapsigargin-regulated factors in MERS-CoV- and SARS-CoV-2-infected cells (Fig. 7c, highlighted in green, orange). Both proteins

are key regulators of two major and highly interconnected intracellular degradation pathways, ERAD and autophagy[45,46], leading us to focus our further analyses of the antiviral effects of thapsigargin on these processes.

The protein HERPUD1 has an essential scaffolding function for the organization of components of the core ERAD complex[47,48]. ERQC and ERAD pathways are critically involved in the qualitative and quantitative control of misfolded or excessively abundant proteins in the ER. If protein folding in the ER fails, the proteins are retro-translocated through a HERPUD1-dependent ER membrane complex to the cytosol for proteasomal degradation[49]. By searching our proteomics data for further ERAD factors, we were able to retrieve a total of 33 (for MERS-CoV) and 20 (for SARS-CoV-2) proteins of the canonical ERQC and ERAD pathways for which a differential expression was observed in virus-infected cells treated with thapsigargin (Fig. 7f). Mapping of these data on the KEGG hsa04141 pathway suggests that thapsigargin enhances or restores these mechanisms at key nodes of ERQC and ERAD in coronavirus-infected cells (Supplementary Fig. 9).

We also intersected the 120 + 63 proteins jointly regulated by thapsigargin in MERS-CoV- and SARS-CoV-2-infected cells with data from a recent genome-wide sgRNA screen that reported new ERAD factors required for protein degradation[50]. This analysis identified 30 additional thapsigargin-regulated factors that may further support antiviral ERAD, including the E1 ubiquitin ligase UBA6 and the zinc finger protein ZNF622 (also called ZPR9), which were recently described either as negative regulators of autophagy or of some DNA virus infections (Fig. 7g)[51,52].

The protein p62/SQSTM1 is a multifunctional signaling protein and cargo receptor that targets clients for destruction by selective autophagy[53]. This raised the question of whether the elevated p62/SQSTM1 levels observed in thapsigargin-treated cells affected autophagy pathways during CoV infection. We, therefore, assessed, under these conditions, viral replication and the autophagic flux by determining the levels of p62/SQSTM1 and the non-lipidated/lipidated forms of the ATG8 ortholog LC3B, a protein that is central to autophagosome formation, in the presence/absence of lysosomal inhibition. Pre-treatment of HuH7 cells with the lysosomal V-ATPase inhibitor bafilomycin $A_1$ for 4 h suppressed HCoV-229E infection to a similar extent as thapsigargin (by about 100-fold) (Fig. 8a, lanes 3 and 4), while the addition of bafilomycin $A_1$ at 8 h p.i. reduced viral replication only 10-fold (Fig. 8a, lanes 5 and 6). Under these conditions, bafilomycin $A_1$ showed no negative effects on cell viability (Supplementary Fig. 10). Bafilomycin $A_1$ alone did not suppress MERS-CoV replication but inhibited SARS-CoV-2 (Fig. 8b).

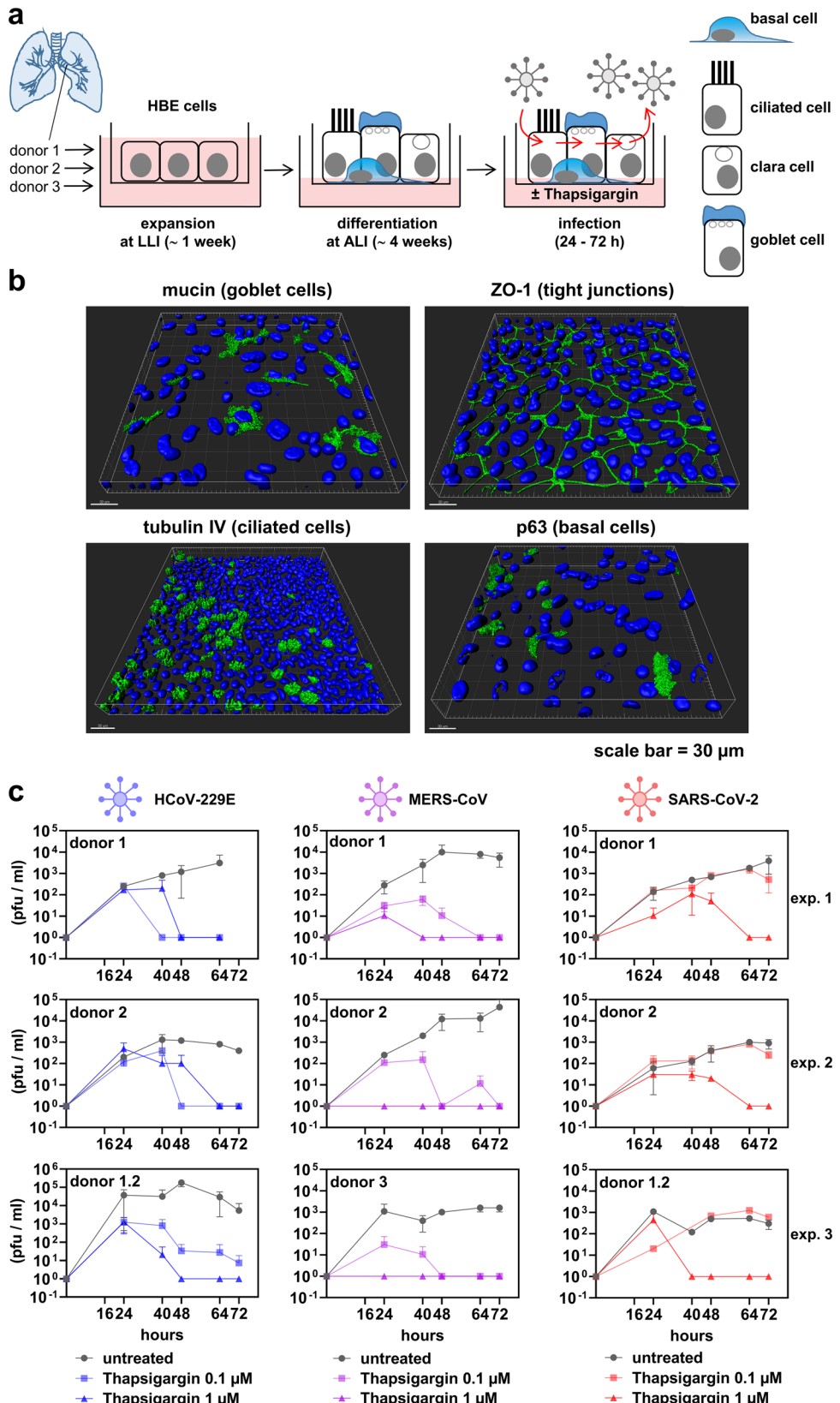

These (variable) antiviral effects of bafilomycin A₁ are in line with other alkylating agents that suppress lysosomal pathways[54].

Bafilomycin A₁ and thapsigargin strongly increased the appearance of p62/SQSTM1-positive foci representing autophagosomes in untreated but also in infected cells, suggesting that HCoV-229E infection but also thapsigargin treatment stimulate early events in autophagosome biogenesis (Fig. 8c).

With respect to viral protein synthesis, pretreatment of cells with bafilomycin A₁ partially reduced HCoV-229E N protein levels but had no effect on viral N protein accumulation when

**Fig. 5 Thapsigargin suppresses CoV replication in differentiated primary human bronchial epithelial cells. a** Scheme showing the expansion in a liquid−liquid interphase (LLI) followed by the differentiation at an air-liquid interphase (ALI) of normal human bronchial epithelial cells (NHBE). **b** Three-dimensional immunofluorescence analysis (z-stacks) of representative NHBE cells stained with antibodies specific for the indicated differentiation markers. Shown is one representative out of two biologically independent experiments. **c** NHBE cells were left untreated or infected with the indicated CoV (MOI = 3) and treated for up to three days with thapsigargin (0.1 or 1 μM). Supernatants were collected at five time points p.i. and virus titers determined by plaque assay. Data represent three biologically independent experiments using NHBE cells derived from two or three independent donors. Shown are means ± s.d. of technical duplicates. For HCoV-229E and MERS-CoV, cells from donor 1 were plated and differentiated a second time to generate an additional independent experiment (labeled donor 1.2). See Supplementary Fig. 5f for cell viability experiments.

added 8 h p.i. This was in clear contrast to thapsigargin, which (similar to the experiments presented above) suppressed N protein accumulation completely (Fig. 8d, e). HCoV-229E infection reduced p62/SQSTM1 levels by approximately 70% (mean 26 ± 8%) (Fig. 8d, e, lane 1 compared with lanes 5 or 14), whereas there was no such effect on total LC3B (mean 105 ± 27%) or lipidated LC3B-II (mean 120 ± 23%) (Fig. 8d, e). Thapsigargin upregulated p62/SQSTM1 (mean 294 ± 30%), total LC3B (mean 196 ± 28%), and LC3B-II (mean 218 ± 27%) (Fig. 8d, e). Blocking lysosome acidification by bafilomycin A₁ strongly induced p62 levels in HCoV-229E-infected cells by six-fold (to a mean of 179 ± 21%) (Fig. 8d, e, lanes 5 and 6) and increased LC3B-II levels by three-fold (mean 321 ± 145%) (Fig. 8d, e, lanes 5 and 6). Under all conditions used, there was no bafilomycin A₁-mediated effect on p62/SQSTM1 or LC3B levels upon addition of thapsigargin (Fig. 8d, e). These changes were used to calculate the turnover of LC3B-II and p62/SQSTM1 as a measure of basal and selective autophagic flux, respectively (Fig. 8f), which were determined according to the procedures described in ref. [55]. Compared to untreated cells, HCoV-229E-infected cells displayed a weakly reduced basal autophagy by about 25%, with selective autophagy being stimulated by around three-fold early upon infection (Fig. 8f). Under all conditions tested, thapsigargin strongly suppressed the autophagic flux (Fig. 8f). These data suggest that the increased levels of p62/SQSTM1 and LC3B-II, respectively, observed in thapsigargin-exposed cells result from a suppression of their lysosomal degradation and lead us to conclude that the blockade of CoV-induced selective autophagic flux represents an additional antiviral mechanism of thapsigargin.

Finally, we sought to validate the protein changes observed by mass spectrometry and by the detailed analysis of the autophagic flux for HCoV-229E, MERS-CoV, and SARS-CoV-2 including also cystathionine-γ-lyase (CTH), an additional hit belonging to the enriched pathways GO:0006520 (cellular amino acid metabolic process) and GO:0034976 (response to ER stress, as shown in Fig. 6g–i). CTH is a regulator of glutathione homeostasis and cell survival[56]. As shown by immunoblotting, we confirmed the upregulation of proteins representative for the UPR (IRE1α, BiP, CTH), for ERQC/ERAD (HERPUD1, UBA6, ZNF622), and for autophagy (p62/SQSTM1, LC3B-I/LC3B-II) in thapsigargin-treated cells infected with one of the three CoVs (Fig. 9a, b).

In conclusion, these data show that thapsigargin forces the (re) expression of a dedicated network of proteins with roles in ER stress, a range of metabolic pathways, ERQC, ERAD, and the regulation of autophagy. Noteworthy, it is the combination of these changes at the protein level that confers a sustained "anti-viral state" and profoundly suppresses CoV replication as summarized schematically in Fig. 9c.

## Discussion

In this study, we report a potent inhibitory effect of the chemical thapsigargin on the replication of three human CoVs in four different cell types. Following up on observations that CoV globally suppresses UPR/ER stress factors, we find that thapsigargin

counteracts the CoV-induced downregulation of BiP, HERPUD1 (and CTH) and increases IRE1α levels. In this context, thapsigargin also plays a role in overcoming the coronavirus-induced block of global protein biosynthesis. Proteome-wide data revealed a thapsigargin-mediated reprogramming of metabolic pathways and helped to identify a network of specific thapsigargin-regulated factors, including candidates from the ERQC/ERAD pathways that, most likely, are involved in the destruction of viral proteins. The positive effects of prolonged thapsigargin treatment on the expression of cellular BiP and HERPUD1 are well documented[57–60]. Thus, one key finding of our study is that the thapsigargin-mediated induction of ER factors overrides suppressive effects of CoVs on ER functions, as illustrated here for BiP, IRE1α and HERPUD1, but also at the global proteomic scale.

BiP is one of the most abundant cellular proteins (also in our mass spectrometry data) and plays an essential role in development and disease[61,62]. In yeast, the inducible expression of the BiP homolog Karp2 was shown to be essential for disposing of toxic proteins and reducing cellular stress[63]. Hence, a reduction of BiP levels (as seen during CoV infection) and the contrary effect of thapsigargin-mediated upregulation are likely to have opposing consequences for the host cell following viral infection. Similarly, IRE1α is suggested to mediate protective and adaptive responses suitable to alleviate ER stress, e.g., by balancing lipid bilayer stress, an aberrant perturbation of ER membrane structures, which may be expected to occur upon DMV formation in CoV-infected cells[18,31,64]. Accordingly, high levels of BiP, HERPUD1, and IRE1α may increase in general the resilience of cells when infected by diverse pathogens. In line with this, our data show that, in cells infected with representative coronaviruses, a protective ER/UPR response is initially elicited at the mRNA level (Fig. 1c and ref. [17]). However, the global suppression at the protein level (or the lack of induction) indicates that CoVs have evolved strategies at the posttranscriptional or translational level to escape the protective antiviral activities of BiP, IRE1α, and HERPUD1.

Together with PERK, all three proteins are key regulators of ERQC/ERAD pathways and there is ample evidence to suggest that their expression, regulation, and activities are intimately linked[21,22]. A recent study reported that PERK activation induces the RPAP2 phosphatase, inactivates IRE1α kinase activity, and aborts IRE1α-mediated adaptive functions in response to the chemical stressor Brefeldin A[65]. In another report, high BiP levels exerted negative control of IRE1α by directly binding the kinase or by promoting IRE1α degradation[66]. Here, we show a different scenario, in which high BiP and IRE1α protein levels coincide with an antiviral state, as well as with improved metabolic functions, suggesting unique modes of cross-regulation of PERK, BiP, and IRE1α in CoV-infected cells exposed to chemical stress.

The (up)regulation of HERPUD1 and several ERAD factors by thapsigargin provides an additional layer of control contributing to the rapid suppression of CoV proteins. While ERAD is generally needed to dispose of unwanted proteins in the ER[67,68], a process called "ERAD tuning" that has been suggested to dampen or balance ERAD activity by

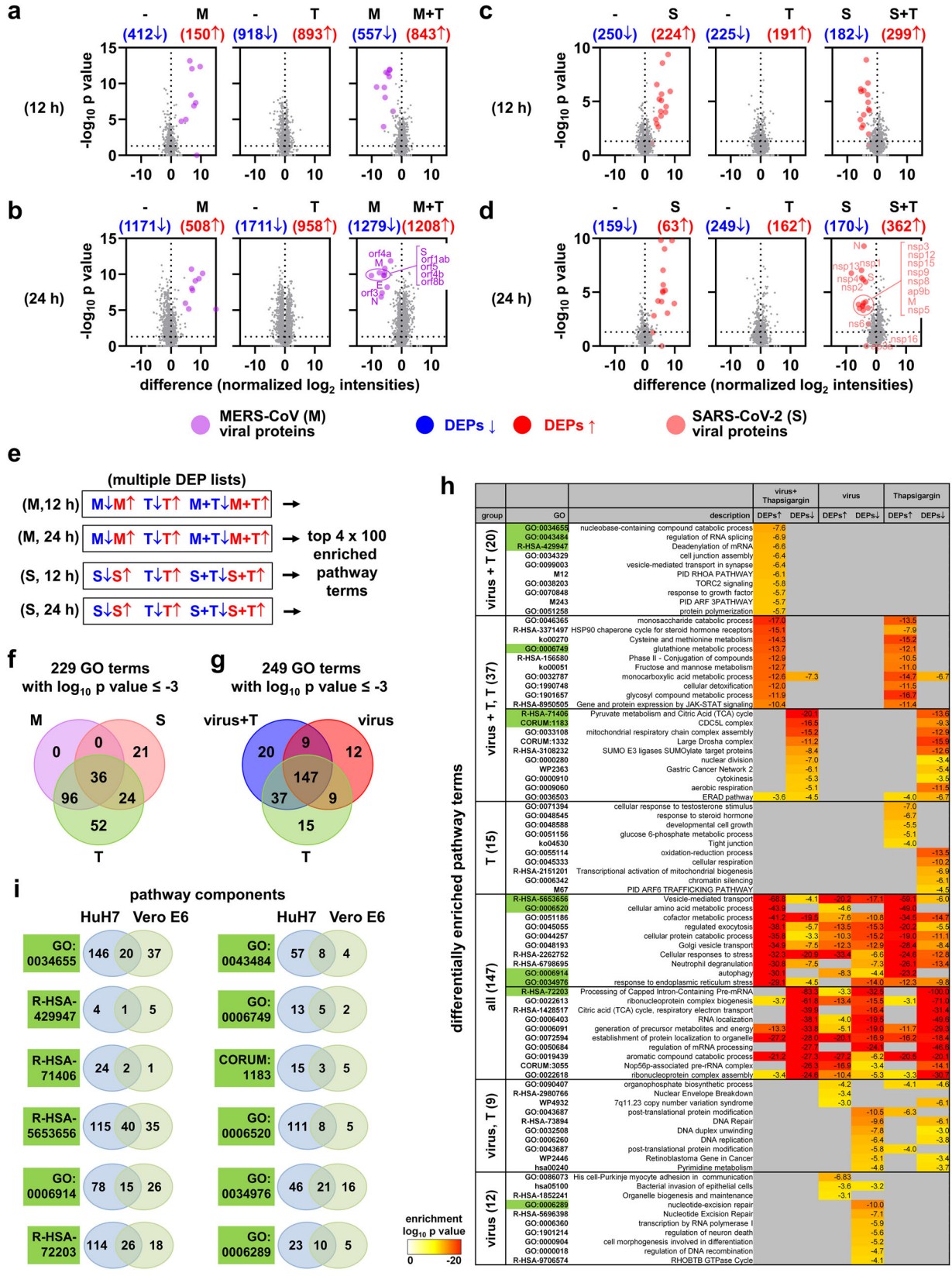

segregating ERAD components into specific ER-derived vesicles called EDEMosomes. During CoV infection, ERAD tuning may prevent the destruction of CoV proteins[69–71]. Our data are compatible with a model in which a modulation of ERAD components by small molecules may antagonize "ERAD tuning"

promoted by CoVs, thereby preserving normal ERAD topology, as well as high ERAD activity.

High HERPUD1 levels were also reported to impair autophagy and, conversely, its depletion increased autophagic flux upon glucose stress[72], suggesting that HERPUD1 may contribute to the

**Fig. 6 Proteome-wide identification of thapsigargin action on MERS-CoV- or SARS-CoV-2-infected cells reveals virus- and thapsigargin-specific protein and pathway patterns.** Total cell extracts from uninfected cells (−), HuH7 cells infected with MERS-CoV (M, MOI = 3) for 12 h (**a**) or 24 h (**b**), or Vero E6 cells infected with SARS-CoV-2 (S, MOI = 3) for 12 h (**c**) or 24 h (**d**), in the presence or absence of thapsigargin (T, 1 µM) were subjected to LC-MS/MS analysis. 5,367 (from HuH7) or 5,066 (from Vero E6 cells) majority protein IDs were identified and their intensities were normalized between samples. Volcano plots show pairwise ratio comparisons and corresponding $p$ values obtained from Student's t-tests, which were derived from the means of two independent experiments for each condition and three technical replicates per sample. Blue and red colors indicate differentially expressed proteins (DEPs, ratio > 0, $p$ value of $-\log_{10}(p) \geq 1.3$). Purple and light red colors indicate individual viral proteins. **e** Majority IDs (for HuH7 cells) or NCBI gene IDs (for Vero E6 cells) corresponding to the DEPs shown in (**a**)−(**d**)) were used for overrepresentation analyses to identify the top 100 enriched pathway categories per virus and time point using Metascape software[44]. Complete lists of pathways are shown in Supplementary Figs. 6 and 7 as clustered heatmaps. The top five enriched pathway categories for up- or downregulated DEPs are shown in Supplementary Fig. 8a. (**f**, **g**) The 400 enriched pathway categories were pooled and filtered for common and distinct pathways considering only terms with enrichment $p$ values of $\log_{10}(p) \leq -3$. **f** Venn diagram showing pathway terms specific to MERS-CoV (M), SARS-CoV-2 (S), or thapsigargin (T). The top 20 enriched pathway categories are shown in Supplementary Fig. 8b. **g** Venn diagram showing pathway terms specific for the virus, thapsigargin, or infection plus thapsigargin (virus + T) conditions. **h** The heatmap shows the top differentially enriched pathways corresponding to the Venn diagram shown in (**g**). Green colors refer to the pathways highlighted in (**i**). **i** Twelve examples of differential and joint gene ID compositions of pathways enriched in HuH7 or Vero E6 cells.

antiviral effects of thapsigargin by a dual-mode, through ERAD and as a repressor of autophagy. The contribution of (macro) autophagy to CoV replication is still controversial with evidence for both pro- and antiviral roles, depending on the virus strains and model systems used (reviewed in refs. [71,73]). Here, we provide evidence to suggest that human CoVs induce (and require) p62/SQSTM1-mediated selective autophagy early in their replication cycle because N protein translation and replication are repressed to some extent by inhibitors of lysosomal acidification. Recently, p62/SQSTM1 was shown to act as an adaptor to mediate a selective form of autophagy, called ER-phagy, that normally serves to remove larger parts of damaged ER or bulky protein aggregates that cannot readily be disposed of by ERAD[74–76]. Based on our findings, we speculate that CoV may leverage ER-phagy, in addition to "ERAD tuning", to generate membrane sources for their specific CM/DMV environment. Compared to lysosomal inhibition, the thapsigargin effects on blocking CoV-mediated autophagic flux were much more sustained and durable. This observation is in line with the work from Ganley et al. who showed that thapsigargin blocks the fusion of autophagosomes with lysosomes, a late step in the autophagy pathway, by an unknown mechanism[77]. Genetic experiments in *Drosophila* revealed a role of the SERCA $Ca^{2+}$ pump in membrane fusion events of lysosomes[78]. Because thapsigargin inhibits SERCA[79], it, therefore, remains a possibility that thapsigargin is particularly efficient in blocking autophagy and CoV-replication because it prevents the $Ca^{2+}$ gradients required for multiple vesicle fusion events known to occur during the CoV replication cycle.

Clearly, the precise mechanistic basis for these effects remains to be identified in additional studies. One approach will be to identify direct interactions between CoV proteins and ER stress or autophagy pathway components. A number of affinity purification- or proximity labeling-based mass spectrometry data sets are now available for host cell proteins interacting with individually expressed SARS-CoV-2 proteins (reviewed in ref. [80]). These data reveal a large number of host interactors from a broad range of pathways, several of which including regulators of ER stress or autophagy[81]. However, it remains to be seen which of the reported protein:protein interactions can be confirmed in CoV-infected cells with their unique subcellular compartmentalization.

In line with the interaction studies, our proteomic and functional data show that thapsigargin affects multiple pathways beyond the core ER stress response. Overall, the available evidence indicates that it will not be trivial to identify the essential targets that mediate thapsigargin's multimodal antiviral effects.

Our data provide a rich resource for further drug target analysis, also in conjunction with the few deep protein sequencing studies available for SARS-CoV-2 (but not MERS-CoV)[82–85]. Thus, our study fills an important knowledge gap by providing a direct side-by-side comparison of pharmacologically targeted cells infected with two highly pathogenic human coronaviruses.

In the absence of specific and effective therapeutic strategies to combat coronaviruses, and in view of the current SARS-CoV-2 pandemic, we hope that our observations will stimulate a broader investigation of this potential therapeutic avenue. This will also include more detailed studies on the antiviral effects and specificity of thapsigargin for other RNA viruses, such as influenza A virus. Given that thapsigargin concentrations in the lower nanomolar range were shown to abolish CoV replication in cultured cells (including primary bronchial epithelial cells) for up to three days after a single application, this work identifies thapsigargin as an interesting drug candidate. The $Ca^{2+}$ mobilizing and cytotoxic features of plant-derived thapsigargin have been studied for 40 years[86,87]. Several analogs have already been designed and efficient and scalable purification or synthesis is now available for application in humans[88–90]. Recently, a protease-cleavable prodrug of thapsigargin, mipsagargin, has been evaluated in phase I and II clinical trials for prostate cancer[86,91–93]. It is not uncommon to adapt anti-proliferative cytostatic drugs (e.g., azathioprine, cylophosphamide, methotrexate) for the treatment of autoimmune and inflammatory disorders by applying lower doses than those needed for treating cancer[94]. Similarly, low doses of thapsigargin combined with short-term systemic or topical application in the airways might reduce viral load early on or in critically ill patients with a favorable therapeutic index with respect to antiviral versus cytotoxic effects. CoVs also activate inflammatory, NF-κB-dependent cytokine and chemokines at the mRNA level[17], some of which (CXCL2, CCL20) escaping translational shut-down and being secreted in a cell-type-specific manner (Supplementary Fig. 11). Some of these cytokines may contribute to the cytokine storm observed in some COVID-19 patients[95]. While thapsigargin had no effect on IL-8, IL-6, CXCL2, and CCL20 in cell culture (Supplementary Fig. 11), a single bolus of the compound was shown to efficiently reduce the translation of pro-inflammatory cytokines in preclinical models of sepsis[96]. Thus, an additional benefit of thapsigargin treatment may arise from dampening overshooting tissue inflammation in COVID-19 patients. In summary, the study provides several lines of evidence that thapsigargin hits a unique combination of central mechanisms required for CoV replication, which may be exploited to develop novel therapeutic strategies. This compound or derivatives thereof with improved specificity, pharmacokinetics, and safety profiles may also turn out to be suitable

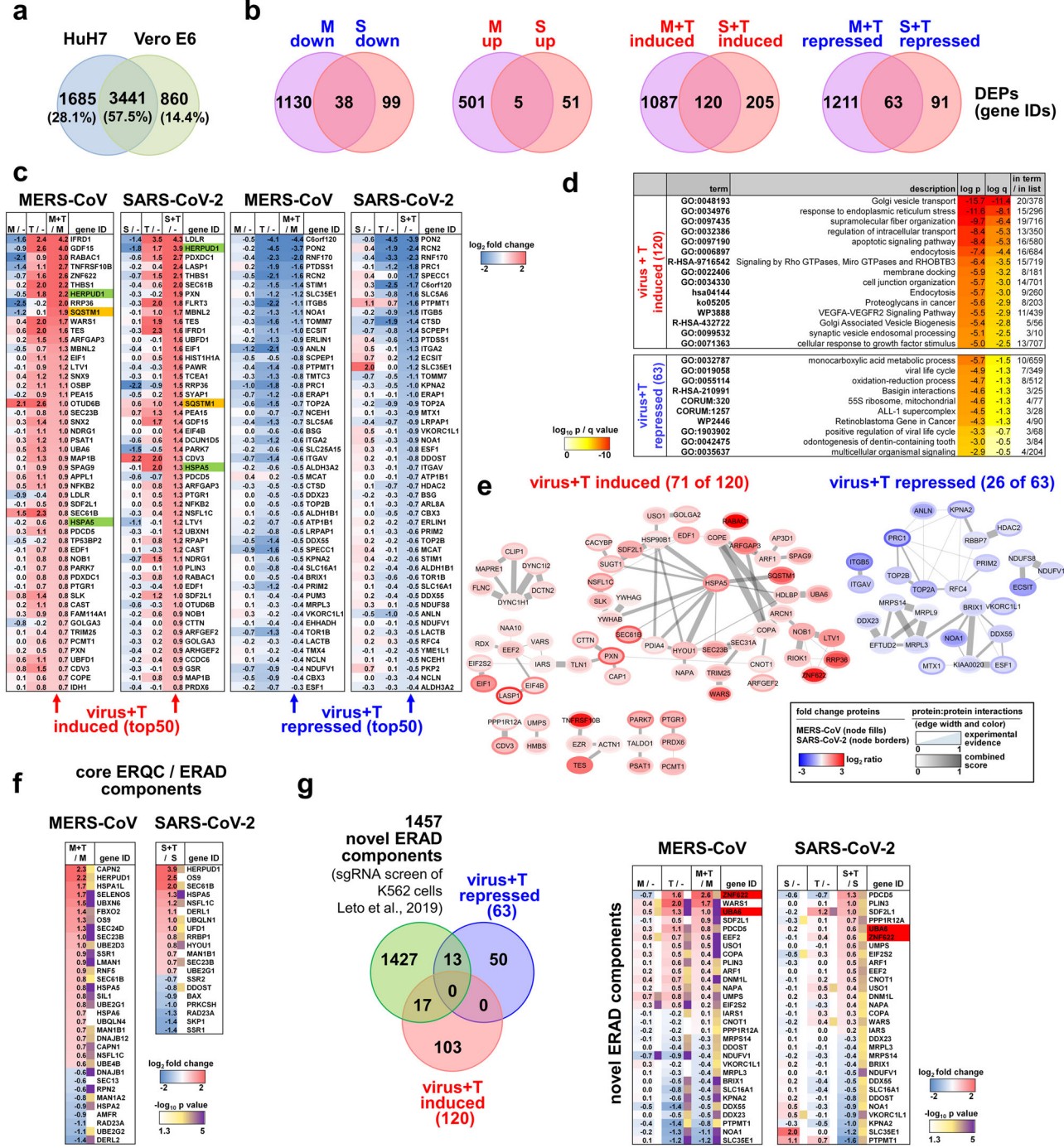

**Fig. 7 Thapsigargin regulates a specific network of proteins involved in transport, ERQC/ERAD, and ER stress in MERS-CoV or SARS-CoV-2-infected cells. a** Overlap of orthologous proteins expressed in HuH7 and Vero cells. **b** Overlap of virus- and thapsigargin-regulated proteins common to HuH7 and Vero E6 cells showing 120 proteins with higher and 63 proteins with lower expression in thapsigargin-treated infected cells compared to virus infection alone (ratio > 0, p value of −log₁₀ (p) ≥ 1.3). **c** Heatmaps showing individual mean ratio values of normalized protein intensities for the top 50 up- or downregulated proteins in virus-infected and thapsigargin-treated cells. Ratio values of infected or thapsigargin-treated conditions compared to untreated cells (−) are shown for comparison and are sorted according to the virus plus thapsigargin conditions. Green colors highlight HERPUD1 and BiP (HSPA5), while orange colors highlight p62/SQSTM1. **d** Top pathways mapping to gene IDs with increased (120 proteins, red) or decreased (63 proteins, blue) expression levels in thapsigargin-treated and infected cells compared to virus infection alone as revealed by overrepresentation analysis using Metascape software[44]. **e** Protein:protein interactions (PPI) amongst the 120 up- and 63 downregulated thapsigargin-sensitive proteins based on experimental evidence, co-occurrence, co-expression, and confidence scores from the STRING database[108]. According to experimental evidence and combined STRING score criteria, 71 and 26 coregulated proteins are engaged in defined PPI networks; the remaining proteins are not known to interact. **f** Heatmap showing thapsigargin-reprogrammed proteins of KEGG hsa04141 (mean ratio ≥ 1.5 fold). See also Supplementary Fig. 9 for projection of thapsigargin-mediated protein changes on the KEGG pathway map. **g** Venn diagram showing the intersection of thapsigargin-/virus-regulated proteins with all novel ERAD components (FDR of 1%) identified by ref. [50]. The regulation of 30 overlapping components is shown as a heatmap displaying mean ratio values in thapsigargin-treated or infected cells. Red colors highlight UBA6 and ZNF622 as discussed in the text. (**b, c, f, g**) Mean ratio and p values were determined as described in the legend of Fig. 6a–d. Abbreviations: *M* MERS-CoV; *S* SARS-CoV-2; *T* thapsigargin.

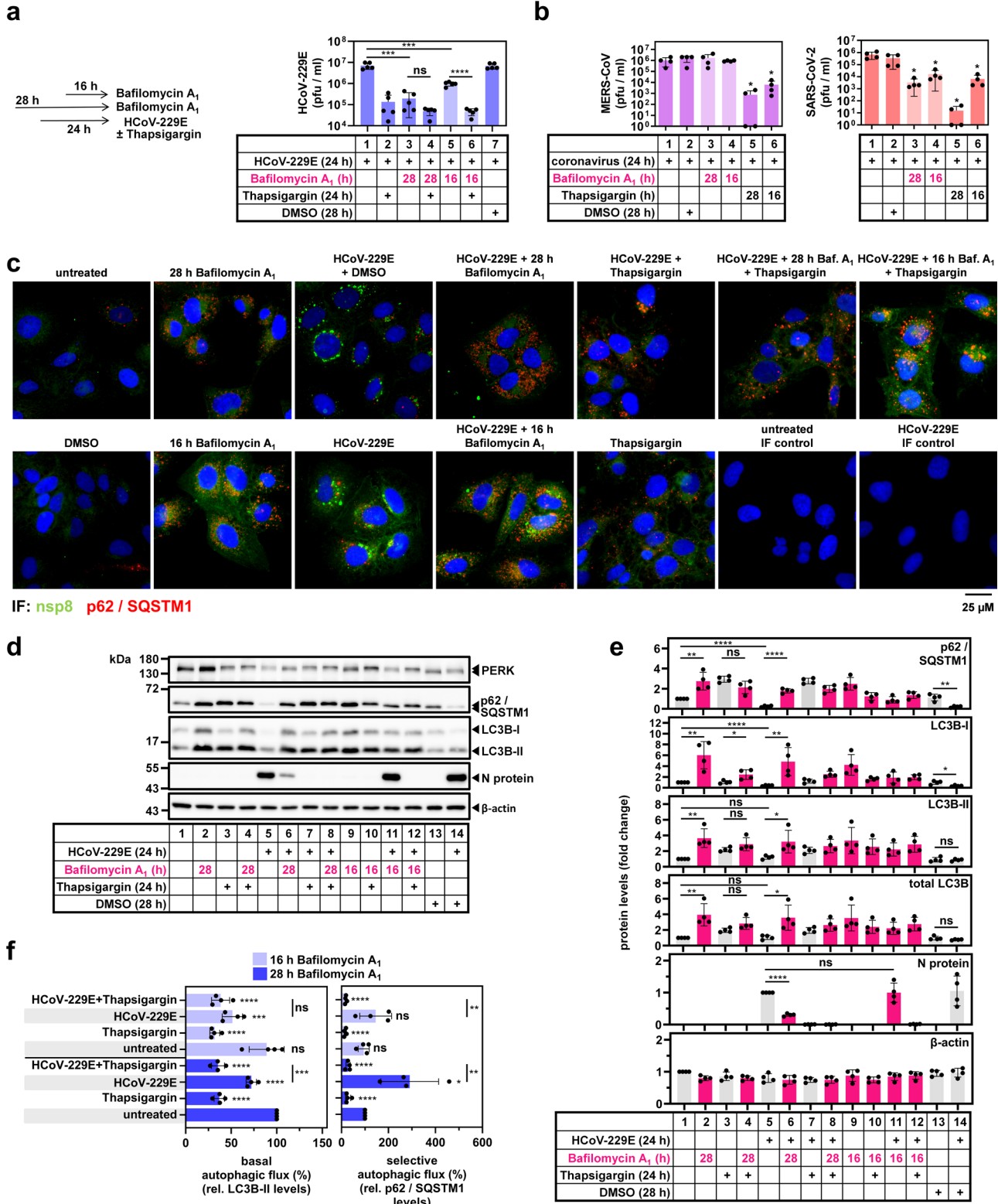

IF: nsp8  p62 / SQSTM1

for mitigating the consequences of potential future CoV epidemics more effectively.

## Methods

**Cells and viruses**. HuH7 human hepatoma cells (Japanese Collection of Research Bioresources (JCRB) cell bank;[97]) were maintained in Dulbecco's modified Eagle's medium (DMEM, including 3.7 g/l NaHCO₃; PAN Biotech Cat No P04-03550) complemented with 10% filtrated bovine serum (FBS Good Forte; PAN Biotech,

Cat No. P40-47500), 2 mM L-glutamine (Gibco, 21935-028), 100 U/ml penicillin and 100 μg /ml streptomycin. MRC-5 human embryonic lung fibroblasts (ATCC, CCL-171) were maintained in DMEM containing 1.5 g/l (w/v) NaHCO₃ and complemented with 10% fetal calf serum (FCS; PAN Biotech Cat No. 1502-P110704), 2 mM L-glutamine, 100 U/ml penicillin, and 100 μg/ml streptomycin, 1% minimum essential medium non-essential amino acids (100x MEM NEAA; Gibco Cat No 11140-035) and 1 mM sodium pyruvate (100 mM; Gibco 11360-039). Vero E6 African green monkey kidney epithelial cells (ATCC CRL-1586), A549 cells (ATCC CCL-185), and MDCK-II (ATCC CRL-2936) cells were grown

**Fig. 8 Thapsigargin suppresses CoV-induced autophagic flux. a** HuH7 cells were treated with bafilomycin A$_1$ 4 h before (28 h) or 8 h after (16 h) infection with HCoV-229E (MOI = 1). Thapsigargin was added as indicated. Samples from DMSO-treated cells served as solvent controls. After 24 h, supernatants were used to determine viral titers (five biologically independent experiments). **b** Effects of pre- or post-infection treatments with bafilomycin A$_1$ as shown in the scheme of (**a**) on viral titers in supernatants collected from HuH7 or Vero E6 cells infected with MERS-CoV (MOI = 0.5, left graphs) or with SARS-CoV-2 (MOI = 0.5, right graphs) for 24 h (four biologically independent experiments). **c** Fluorescence microscopy images representative for one out of two biologically independent experiments showing the distribution and subcellular localization of p62/SQSTM1-positive foci in untreated cells or cells infected with HCoV-229E in the presence/absence of bafilomycin A$_1$ and thapsigargin. Viral replication sites were stained with antibodies specific for nsp8. **d, e** HuH7 cells were treated or infected with HCoV-229E as described in (**a**). Total cell extracts were analyzed by immunoblotting for PERK, the non-lipidated (LC3B-I) or lipidated (LC3B-II) forms of LC3B, p62/SQSTM1, and N protein as indicated. Antibodies against β-actin were used to validate equal loading. **d** Shows a representative experiment and **e** shows the quantification of immunoblot data from four independent experiments relative to the untreated control. **f** Data from (**d, e**) were used to calculate the autophagic flux relative to untreated and non-infected conditions. All bar graphs show means ± s.d.; asterisks indicate *p* values (*$p \leq 0.05$, **$p \leq 0.01$, ***$p \leq 0.001$, ****$p \leq 0.0001$) obtained by two-tailed unpaired t-tests. See Supplementary Fig. 10 for cell viability assays under bafilomycin A$_1$ treatment conditions.

in DMEM, 10% FBS, 100 U/ml penicillin, and 100 µg/ml streptomycin. HuH7 and MRC-5 cells were confirmed to be free of mycoplasma using the Venor® GeM Classic kit (Minerva Biolabs).

Cryopreserved normal human bronchial epithelial (NHBE) cells were obtained from Lonza. All donors (donor 1: TAN 24717, Lot No. 000312626; donor 2: TAN 36585, Batch: 18TL269120; donor 3: TAN 28968, Lot No. 00004859609) were non-smoking and lacking respiratory pathology. Undifferentiated cells were seeded on transwell plates (Corning Costar, CLS3470-48EA) coated with collagen IV (Invitrogen) and grown in a mixture of DMEM (Invitrogen) and BEGM (Lonza, CC-3170) supplemented with retinoic acid (75 nM). Fresh medium was added regularly after 2 days. After reaching confluence, the cells were cultivated under air-liquid conditions for four additional weeks until full differentiation into pseudostratified human airway epithelia was observed. Medium from the basolateral compartment was renewed every 2–3 days and the apical surface was washed every week with PBS (Invitrogen).

Genome sequences of coronavirus strains used in this study are as follows: HCoV-229E (NCBI accession number AF304460.1, NCBI reference sequence NC_002645.1), MERS-CoV (NCBI accession number JX869059, NCBI reference sequence NC_019843.3). SARS-CoV-2 genome sequencing data have been submitted to the NCBI Short Read Archive repository under bioproject PRJNA658242 (SRA accession number SRX9907172 and SRX8975039). MERS-CoV and SARS-CoV-2 were kindly provided by Christian Drosten. For infection with other RNA viruses, influenza A virus (A/Thailand/1(KAN-1)/2004(H5N1); IAV; NCBI:txid266827) and human poliovirus type 1 (strain Mahoney, NCBI:txid12081) were used.

**Virus infections and assessments of antiviral activity.** To analyze the antiviral activity of thapsigargin, HuH7 cells (for HCoV-229E, MERS-CoV), MRC-5 cells (for HCoV-229E), Vero E6 cells (for SARS-CoV-2, poliovirus), and A549 cells (for IAV), respectively, were infected at the indicated multiplicities of infection (MOI) and incubated at 33 °C (for HCoV-229E and SARS-CoV-2) or 37 °C (for MERS-CoV, IAV, poliovirus) in the presence or absence of thapsigargin, or with the appropriate volume of solvent control (DMSO) as indicated. At 24 h p.i., supernatants were collected and stored at −80 °C. Virus titers in the supernatants were determined by plaque assay. HCoV-229E was titrated on HuH7 cells seeded on 12-well plates. MERS-CoV was titrated on HuH7 cells seeded on 24-well plates, SARS-CoV-2 and poliovirus were titrated on Vero E6 cells seeded on 24-well plates and IAV was titrated on MDCK-II cells seeded on 24-well plates using standard procedures. Briefly, confluent monolayers of the appropriate cells were incubated with serial dilutions of virus-containing supernatants (diluted $10^1$ to $10^7$) and incubated at 33 °C (HCoV-229E, SARS-CoV-2) or 37 °C (MERS-CoV, poliovirus, IAV). After 1 h, the virus inoculum was replaced with fresh medium (MEM, Gibco) containing 1.25% Avicel® (FMC Biopolymer, #RC591), 100 U/ml penicillin, 100 µg/ml streptomycin, and 10% FBS. For IAV infection, FBS was replaced with 0.2% bovine albumin (Sigma, A7979) and 1 µg/ml TPCK-treated trypsin). For poliovirus infection, 40 mM of MgCl$_2$ was added to the titration medium. At 48 h p.i. (for MERS-CoV, IAV, PV) or 72 h p.i. (for SARS-CoV-2 and HCoV-229E), cell culture supernatants were removed. Cells were washed with PBS and fixed in freshly prepared 3.7% PFA in PBS overnight. Next, the fixing solution was removed, the cell layer was washed with PBS and stained with 0.15% (w/v) crystal violet (diluted in 20% Ethanol) and plaques were counted. For EC$_{50}$ calculation, virus titers determined for virus-infected cells treated with DMSO only (no inhibitor) were used for normalization. EC$_{50}$ values were calculated by non-linear regression analysis using GraphPad Prism 5.0 or 8.4.3 (GraphPad Software). All virus work was performed in biosafety level 2 (BSL-2; HCoV-229E) or biosafety level 3 (BSL-3; MERS-CoV, SARS-CoV-2, IAV, poliovirus) containment laboratories approved for such use by the local authorities (RP Giessen, Germany).

For the infection of the NHBE cells, the apical surface was washed three times with PBS, and cells were infected with the indicated virus (MOI = 3). After 1 h, the inoculum was removed and the medium in the basal compartment was replaced with a medium containing 1, 0.1, or 0 µM thapsigargin. At the indicated time

points, the apical surface of the cells was incubated with 150 µl/well PBS for 15 min and virus titers in the supernatants were determined by plaque assay.

**Materials.** Thapsigargin (Cay10522-1), GSK2656157 (Cay17372), and remdesivir (Cay30354) were obtained from Cayman Chemicals and were dissolved in DMSO at a 10 mM stock solution. Thapsigargin was used at 1 µM concentration unless stated otherwise. Bafilomycin A$_1$ (Cay11038) was dissolved as 1 mM stock in DMSO and used at 1 µM in all assays. Appropriate DMSO concentrations served as vehicle controls in some experiments. The following inhibitors were used: leupeptin hemisulfate (Carl Roth, #CN33.2), microcystin (Enzo Life Sciences, #ALX-350012-M001), pepstatin A (Applichem, #A2205), PMSF (SigmaAldrich, #P-7626). Pepstatin A, PMSF, and microcystin were dissolved in ethanol and leupeptin in water. Other reagents were from Sigma-Aldrich or Thermo Fisher Scientific, Santa Cruz Biotechnology, Jackson ImmunoResearch, or InvivoGen and were of analytical grade or better.

Primary antibodies against the following proteins or peptides were used: anti β-actin (Santa Cruz, #sc-47778, 1:1000), anti PERK (Santa Cruz, #sc-377400, 1:1000), anti PERK (Abcam, #ab65142, 1:1000), anti BiP (Cell Signaling, #3177, 1:1000), anti eIF2α (Cell Signaling #9722, 1:1000), anti P(S51)-eIF2α (Cell Signaling #9721, 1:1000), anti P(S724)-IRE1α (Novus Biologicals, #NB100-2323, 1:1000), anti IRE1α (Santa Cruz, #sc-390960, 1:1000), anti ATF4 (Santa Cruz, #sc-390063, 1:1000), anti ATF3 (Santa Cruz, #sc-188, 1:500), anti HERPUD1 (Abnova, #H00009709-A01, 1:1000), anti CTH (Cruz, #sc-374249, 1:1000), anti HCoV-229E N protein (Ingenasa, Batch 250609, 1:500), rabbit anti HCoV-229E nsp12 (Eurogentec; directed against full-length nsp12 produced in *E. coli* and purified by the Ziebuhr laboratory, 1:500), rabbit anti HCoV-229E nsp8 ([98] 1:500, IF 1:100), anti MERS-CoV N protein (Sinobiological, #100213-RP02, 1:1000), rabbit anti SARS-CoV N protein cross-reacting with SARS-CoV-2 N protein (gift from Friedemann Weber,[99] 1:2000), anti SARS-CoV-2 N protein (Rockland, #200-401-A50, 1:2000), anti puromycin (Kerafast Inc., 3RH11, #EQ 0001, 1:1000), anti ds-RNA J2 (SCICONS, English & Scientific Consulting Kft, #10010200, IF 1:100), anti p62/SQSTM1 (Santa Cruz, #sc-28359, 1:500, IF 1:100), anti ZNF622 (ZPR9) (Bethyl Laboratories, #A304-076A, 1:1000), anti LC3B XP® (Cell signaling, #3868, 1:1000), anti UBA6 (Cell signaling, #13386, 1:1000), anti ZO-1 (Invitrogen, #40-2200, IF 1:100), anti p63 (abcam, #ab124762, IF 1:100), anti mucin 5AC (abcam, #ab198294, IF 1:100), anti tubulin IV (abcam, #ab179509, 1:100).

The following secondary antibodies were used: polyclonal goat anti mouse immunoglobulins/HRP (Dako P0447, 1:2000), polyclonal goat anti rabbit immunoglobulins/HRP (Dako, P0448, 1:2000), Cy3-coupled anti rabbit (rb) IgG (dk, Merck Millipore, #AP182C, IF 1:200), Dylight 488-coupled anti mouse (ms) IgG (dk, ImmunoReagent, #DkxMu-003D488NHSX, IF 1:200), Alexa Flour594-coupled goat anti mouse IgG (H + L) (Invitrogen, #A11005, IF 1:100), Alexa Fluor 488-coupled F(ab')2 goat anti rabbit IgG (H + L) (Invitrogen, #A-11070, IF 1:100).

**Cell lysis, in vivo puromycinylation, and immunoblotting.** For whole-cell extracts, samples derived from experiments performed with HCoV-229E were lysed in Triton cell lysis buffer (10 mM Tris, pH 7.05, 30 mM NaPPi, 50 mM NaCl, 1% Triton X-100, 2 mM Na$_3$VO$_4$, 50 mM NaF, 20 mM ß-glycerophosphate and freshly added 0.5 mM PMSF, 2.5 µg/ml leupeptin, 1.0 µg/ml pepstatin, 1 µM microcystin). After 10−15 min on ice, lysates were cleared by centrifugation at 15,000 × *g* for 15 min at 4 °C. Protein concentrations of supernatants were determined by Bradford assay and samples stored at −80 °C.

To label nascent polypeptides in intact cells,[100] HuH7 cells were seeded in 6 cm cell culture dishes (3 × $10^5$ cells) and treated as described in the figure legends. Thirty minutes prior to harvest, the medium was supplemented with 3 µM puromycin (InvivoGen, #ant-pr-1). Then, cells were lysed as described above. After immunoblotting (see below), membranes were stained with Coomassie brilliant blue and then hybridized with an anti puromycin antibody (Kerafast, #EQ0001) to detect puromycinylated polypetides.

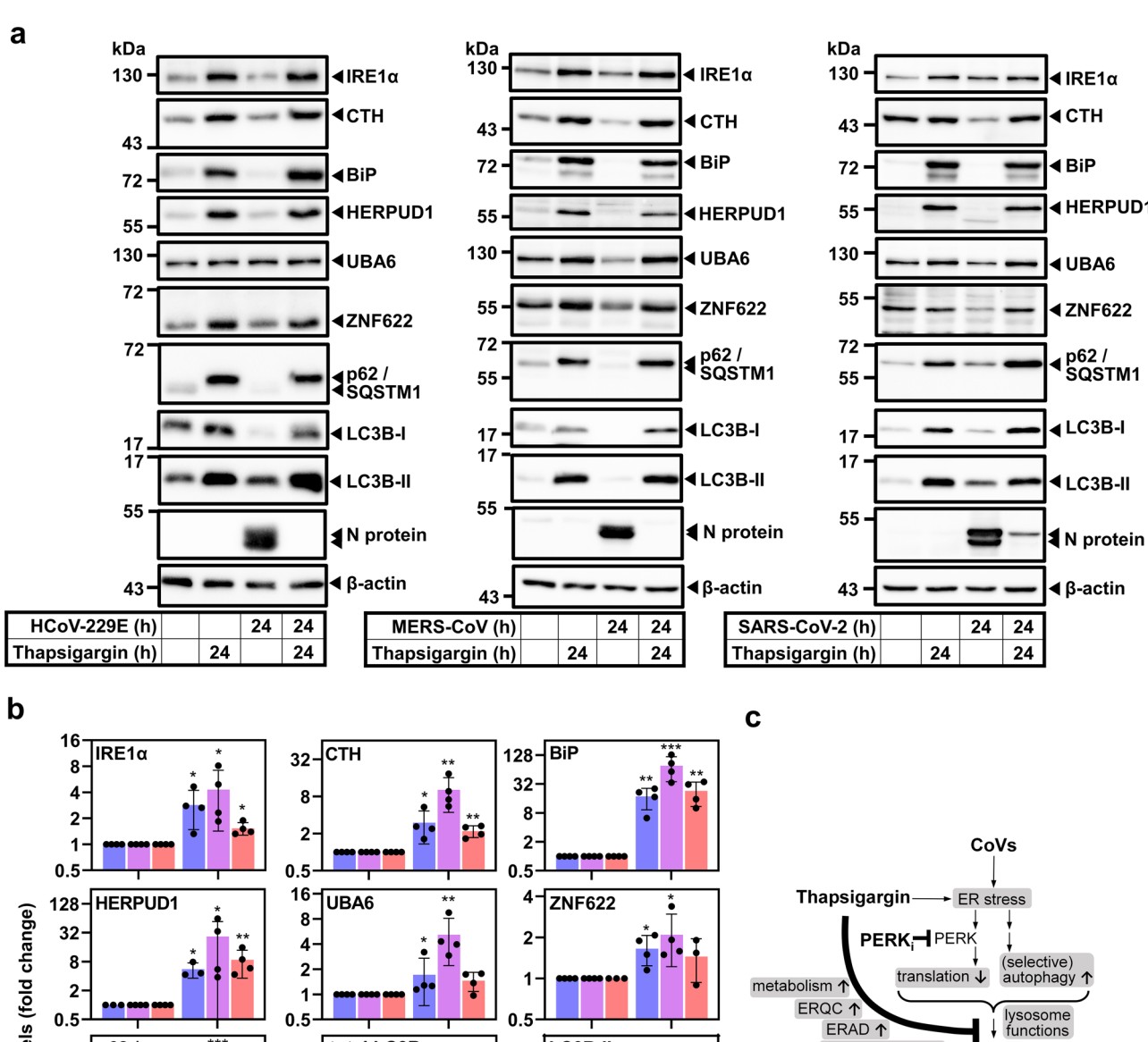

**Fig. 9 Thapsigargin induces key regulators of UPR, ERQC, ERAD, and autophagy in HCoV-229E, MERS-CoV- and SARS-CoV-2-infected cells.**
**a**, **b** Validation of thapsigargin-induced proteins in CoV-infected HuH7 or Vero E6 cells by immunoblotting of whole-cell extracts from cells treated as indicated. BiP and IRE1α levels are shown for comparison. **a** Depicts representative images and **b** shows quantification of four biologically independent experiments, except for HERPUD1 (HCoV-229E samples) or LC3B, LC3B-I, and LC3B-II levels from MERS-CoV-infected cells which were quantified from three experiments. Bar graphs show means ± s.d.; asterisks indicate *p* values (*$p \leq 0.05$, **$p \leq 0.01$, ***$p \leq 0.001$, ****$p \leq 0.0001$) obtained by two-tailed ratio-paired or Mann−Whitney t-tests. Note that LC3B-I and LC3B-II images represent different exposures of the same blot membrane (see source files). **c** Summary of the main findings of our study.

Total cell lysates of MERS-CoV- and SARS-CoV-2-infected cells used for immunoblotting or mass spectrometry were prepared as follows. Cells were scraped in ice-cold PBS and pelleted at $500 \times g$ for 5 min at 4 °C. Cell pellets were washed in ice-cold PBS and stored in liquid $N_2$ (or lysed and processed immediately). After thawing, cell pellets (corresponding to ≈300.000 cells seeded in 60 mm dishes at the start of the experiment) were resuspended in 90 μl of ice-cold $Ca^{2+}/Mg^{2+}$-free PBS and transferred to fresh tubes. After the addition of 10 μl of 10% SDS, samples were heated at 100 °C for 10 min and centrifuged at $600 \times g$ for 1 min at room temperature. Supernatants were transferred to a fresh tube and heated again at 100 °C for 10 min and centrifuged at $600 \times g$ for 1 min at room temperature. Protein concentrations were determined with the detergent compatible Bradford assay kit (Pierce™, #23246) using a 150-fold dilution. Aliquots corresponding to 20−25 μg protein (per lane) were mixed with 4 × SDS sample buffer (ROTI®Load, Roth, #K929) and stored at −20 °C prior to SDS-PAGE, or loaded immediately. Cell lysates were subjected to SDS-PAGE on 8−12.5% gels. The PageRuler™ prestained protein ladder (Thermo Scientific, #26616) was used as a molecular weight marker.

For immunoblotting, proteins were separated on SDS-PAGE and electrophoretically transferred to PVDF membranes (Roti-PVDF, #T830 Roth). Membranes were stained with 0.1% (w/v) Ponceau S (Sigma) dissolved in 5% acetic acid or Coomassie brilliant blue to confirm the transfer and equal loading of proteins. After blocking with 5% dried milk in Tris-HCl-buffered saline/0.05% Tween (TBST) for 1 h, membranes were incubated for 12−24 h with primary antibodies, washed in TBST, and incubated for 1−2 h with the peroxidase-coupled secondary antibody. Proteins were detected by using enhanced chemiluminescence (ECL) systems from Millipore or GE Healthcare. Images were acquired with the ChemiDoc TouchImaging System (BioRad) and quantified using the software ImageLab (versions V_5.2.1 or V_6.0.1, Bio-Rad).

**mRNA expression analysis by RT-qPCR.** 0.25−1 μg of total RNA was prepared by column purification (Macherey-Nagel) and transcribed into cDNA using Moloney murine leukemia virus reverse transcriptase (RevertAid Reverse Transcriptase, Thermo Fisher Scientific, #EP0441) in a total volume of 20 μl. 1 or 2 μl of this reaction mixture was used to amplify cDNAs using Taqman assays on demand (0.25 or 0.5 μl) (Applied Biosystems/Thermo Fisher Scientific) for *GUSB* (81 bp, Hs99999908_m1), *IL6* (95 bp, Hs00174131_m1), *IL8* (101 bp, Hs00174103_m1), *CXCL2* (68 bp, Hs00236966_m1), and *CCL20* (81 bp, Hs00171125_m1), as well as TaqMan Fast Universal PCR Master Mix (Applied Biosystems/ Thermo Fisher Scientific). Alternatively, primer pairs were designed and used with 2 μl of cDNA and Fast SYBR Green PCR Master Mix (Applied Biosystems/Thermo Fisher Scientific) to detect the mRNA of *EIF2AK3* (encoding PERK) (fw5′-AGAGATTGA GACTGCGTGGC-3′, re 5′-TCCCAAATACCTCTGGTTTGCT-3′), nsp8 RNA (fw 5′-GCTGTTGCAAATGGTTCCTCAC-3′, re 5′-GATGCACATTCTTACCATC ATTATCC-3′) and of HCoV-229E S RNA (fw5′-TTTCAGGTGATGCTCACA TACC-3′, re 5′-ACAAACTCACGAACTGTCTTAGG-3′). All PCRs were performed in duplicate on an ABI 7500 real-time PCR instrument. The cycle threshold value (ct) for each individual PCR product was calculated by the instrument's software, and the ct values obtained for inflammatory/target mRNAs were normalized by subtracting the ct values obtained for GUSB. The resulting Δct values were also used to calculate relative changes of mRNA expression as the ratio ($R$) of mRNA expression of treated/untreated cells according to the following equation: $R = 2^{-((\Delta ct\ treated) - (\Delta ct\ untreated))}$. Alternatively, S RNA and nsp8 RNA copy numbers were determined using absolute quantification against a standard curve derived from gel-purified RT-PCR products of nsp8 and S RNA.

A list of oligonucleotides is provided in Supplementary Table 1.

**Immunofluorescence.** Cells were seeded in μ-slides VI (Ibidi) and pre-cultured at 37 °C, 6% $CO_2$. Virus infection as well as simultaneous thapsigargin treatment (1 μM) was performed for 24 h at 33 °C, 6% $CO_2$. After 2× washing, cells were fixed with 4% paraformaldehyde in PBS (Santa Cruz, #281692) for 5 min, washed 3 × 10 min with Hank's BSS (PAN, #P04-32505), blocked with 10% normal donkey serum (Jackson ImmunoResearch, #017-000-121) for 20 min and incubated with primary and secondary antibodies diluted in Hank's BSS containing 0.005% saponin (Sigma-Aldrich, #S4521-10G) for 2 h at room temperature. Following three washing steps with Hank's BSS containing 0.005% saponin, Cy3-conjugated (Millipore, #AP182C, 1:100), Dylight488-conjugated (ImmunoReagents #DkxMu-003D488NHSX, 1:100) secondary antibodies or Alexa Fluor 594 goat anti-mouse IgG (Invitrogen, A11005) and Alexa Fluor 488 F(ab)2-goat anti-rabbit IgG (Invitrogen, A11070) were used. For controls, primary antibodies were omitted. Nuclei were stained with Hoechst 33342 (Invitrogen). Immunofluorescence images were analyzed using Leica DMi8, Leica SP05, and the Leica LASX software. For double immunofluorescence analyses appropriate filter cubes were used (Dylight488: excitation 480/40 and emission 527/30, Cy3: excitation 560/40 and emission 630/75, as well as counterstaining with Hoechst 33342: excitation 405/60 and emission 470/50). For three-dimensional reconstructions, z-stacks were analyzed using Imaris software (Bitplane, version 8.4).

**Cell viability assays.** MTS assays of HCoV-229E experiments were performed using the The CellTiter 96® AQueous One Solution Cell Proliferation Assay kit

(Promega, #G3582). In brief, $1.2 \times 10^4$ HuH7 or $1 \times 10^4$ MRC-5 cells were seeded in 96-well plates for 24 h and thereafter treated with DMSO, thapsigargin, remdesivir, GSK2656157, bafilomycin $A_1$, virus alone or virus plus chemical for 16, 24, or 28 h as indicated in the figure legends. Then, the medium was replaced by 100 μl complete cell culture medium including 4 μl or 20 μl CellTiter 96® AQueous one solution reagent according to the manufacturer's recommendations. Cells were further incubated for 1 h at 33 °C. Then, absorbance values were measured at 490 nm. Control wells containing only medium and reagent were used to correct for background absorbance. Relative values for cell viability were calculated in relation to the mean of all untreated controls (set to 100%).

For MTT and ATPlite assay (Perkin Elmer), Vero E6 cells seeded at near confluency were incubated with a serial dilution of thapsigargin in a 96-well format. After 24 h, either 200 μl MTT mix (DMEM supplemented with 10% FCS containing 250 μg/ml tetrazolium bromide, Sigma) or 100 μl ATPlite assay buffer was added to each well. For ATPlite assay, cells were incubated for 10 min and luminescence was measured using Spark 10 M (Tecan). For MTT assay, cells were incubated for 90−120 min at 37 °C and fixed using 3.7% PFA in PBS. The tetrazolium crystals were dissolved by adding 200 μl/well isopropanol and the absorbance at 490 nm was measured using an ELISA reader (BioTek). To determine $CC_{50}$ values, the MTT/ATPlite values were calculated in relation to the untreated control (which was set to 100%). $CC_{50}$ values were then calculated by non-linear regression using GraphPadPrism 5.0 (GraphPad Software).

For analyzing cytotoxicity in NHBE cells after 72 h treatment, the transepithelial electrical resistance (TEER) was measured using Epithelial Volt/Ohm Meter 3 (EVOM3, WPI). The obtained TEER values were compared to those obtained for untreated cells.

**ELISA.** Sandwich ELISAs from R&D Systems (DuoSet ELISA for human IL-8 (DY208), IL-6 (DY206), CXCL2 (DY276-05), CCL20 (DY360)) were used to measure secreted human cytokine /chemokine protein concentrations in cell culture supernatants of HuH7 or MRC-5 cells treated as described in the figure legends. The cell culture supernatants were harvested, centrifuged at $15,000 \times g$ at 4 °C for 15 s, and stored at −80 °C. 100 μl of the supernatants were either used undiluted or were diluted in cell culture medium as follows (HuH7: IL-8 (1:10), CXCL2 (1:3), CCL20 (1:8), MRC-5: IL-8 (1:10), IL-6 (1:20), CXCL2 (1:1.5)) and ELISAs were performed according to the manufacturer's instructions using serial dilutions of recombinant proteins as standards. All measurements were within the linear range of the standard curve. In some experiments, an IL-1α (10 ng/ml) stimulation for 16 h was used as a positive control.

**RNA-seq and bioinformatics.** For the data shown in Fig. 1, total RNA was isolated from uninfected and infected cells obtained at 3, 6, 12, 24 h p.i. (or mock infection) using two biological replicates resulting in 32 RNA-seq data sets. RNA was sequenced (with rRNA depletion) using Illumina reagents and an Illumina HiSeq 4000 instrument (single read, 150 bases). Quality control of RNA-seq reads was performed using the FastQC (https://www.bioinformatics.babraham.ac.uk/projects/fastqc/) command line tool version 0.11.7.

After adapter trimming using fastp version 0.19.7[101], the reads were aligned to an index based on human genome hg38 using STAR version 2.7.0d[102].

Gene-specific read counts based on hg38 UCSC gene annotations were extracted using FeatureCounts from the R Subread package version 1.6.3[103] and were imported into R, versions 3.4.4 to 3.6.2 (R Core Team, 2018, https://www.R-project.org/).

Detection of differentially expressed genes was done using DESeq2 version 1.22.1[104]. For significance testing, DESeq2 utilizes the Wald test with adjustments for multiple testing by the Benjamini and Hochberg procedure[104]. From the entire data set, only normalized read counts and ratio values for 166 gene IDs assigned to KEGG hsa04141 were extracted and further analyzed.

**Mass spectrometry and bioinformatics.** Protein extracts were lysed in SDS lysis buffer as described above. Prior to digestion, the SDS-containing solution was exchanged to 8 M urea applying the filter-aided sample preparation (FASP) for proteome analysis protocol using Microcon YM-30 filter devices (Millipore, Cat. MRCF0R030)[105]. Cysteines were alkylated with Iodoacetamide and 8 M urea buffer was exchanged to 50 mM ammonium-bicarbonate buffer with a pH of 8.0. Samples were digested within the filter devices by the addition of sequencing grade modified trypsin (Serva) and incubated at 37 °C over-night. Thereafter, the filter-units were transferred to fresh tubes. Peptides were eluted by the addition of 50 μL 0.5 M NaCl solution and centrifugation ($14.000 \times g$ for 10 min). After drying the filtrates in a vacuum concentrator (Speed Vac), pellets were resuspended in 25 μL of 0.1% formic acid.

Peptides were desalted and concentrated using Chromabond C18WP spin columns (Macherey-Nagel, Part No. 730522). Finally, peptides were dissolved in 25 μl water with 5% acetonitrile and 0.1% formic acid. The mass spectrometric analysis of the samples was performed using a timsTOF Pro mass spectrometer (Bruker Daltonic). A nanoElute HPLC system (Bruker Daltonics), equipped with an Aurora C18 RP column (25 cm × 75 μm) filled with 1.7 μm beads (IonOpticks), was connected online to the mass spectrometer. A portion of approximately 200 ng of peptides corresponding to 2 μl was injected directly on the separation column.

Sample loading was performed at a constant pressure of 800 bar. Separation of the tryptic peptides was achieved at 50 °C column temperature with the following gradient of water/0.1% formic acid (solvent A) and acetonitrile/0.1% formic acid (solvent B) at a flow rate of 400 nl/min: Linear increase from 2%B to 17%B within 60 min, followed by a linear gradient to 25%B within 30 min and linear increase to 37% solvent B in additional 10 min. Finally, B was increased to 95% within 10 min and hold for an additional 10 min. The built-in "DDA PASEF-standard_1.1sec_cycletime" method developed by Bruker Daltonics was used for mass spectrometric measurement. Data analysis was performed using MaxQuant with the Andromeda search engine and Uniprot databases were used for annotating and assigning protein identifiers[106]. Perseus software (versions 1.6.10.50 for HuH7 and 1.6.14.0 for VeroE6 proteomes) was used for further analyses[107].

For the data shown in Fig. 1, raw data from 47 LC-MS/MS runs (representing two independent experiments and three technical replicates per sample for the 3, 6, 12, 24 h infection time points with the exception of the MERS-CoV 24 h time point which has only five replicates, were mapped to the manually annotated and reviewed *Homo sapiens* proteome (UniProtKB/Swiss-Prot Release 2019_06 of 03-Jul-2019), log$_2$-transformed and normalized using the width adjustment function of Perseus. The expression values assigned to uninfected HuH7 cells were derived from a total of 59 mock samples representing multiple technical repeats of two biological samples generated at each of the 3, 6, 12, 24 h time points in order to generate a common reference sample for the mean protein expression found in uninfected/untreated HuH7 cells. This mean reference was used to calculate all pairwise ratio values. The significance of changes was tested by the Student's t-test with a FDR of 5% (250 permutations) using Perseus functions. From the entire data set, only protein intensity values for 166 uniprot IDs assigned to KEGG hsa04141 were extracted and further analyzed using the software tools described below.

For the data shown in Figs. 6 and 7, raw data from 96 LC-MS/MS runs (representing two independent experiments per time point and three technical replicates per sample) were mapped to *Homo sapiens* (uniprot ID UP000005640 for HuH7 cells), *Chlorocebus sabaeus* (Green monkey, *Cercopithecus sabaeus*, uniprot ID UP000029965 for Vero E6 cells), MERS-CoV (uniprot IDs UP000139997 and UP000171868) or SARS-CoV-2 (uniprot ID UP000464024) peptide sequences. All data sets were processed by MaxQuant version 1.6.10.43 (raw data submission was done with version 1.6.17.0)[106] including the match between runs option enabled resulting in the identification of 5,376 (HuH7 cells) or 5,066 (Vero E6 cells) majority protein IDs. For further quantifications, log$_2$-transformed protein intensities were width normalized with Perseus[107] and IDs assigned to contaminants, and reverse sequences were omitted resulting in data sets of 5,172 protein IDs (assigned to 5,130 gene IDs) for HuH7 or 4,873 protein IDs (assigned to 4,305 gene IDs) for Vero E6 cells. For calculation of ratio values between conditions, the 2 × 3 replicates from each condition were assigned to one analysis group. DEPs were identified from log$_2$ transformed normalized protein intensity values by Student's t-test analysis using Perseus functions. Subsequent filtering steps and heatmap representations were performed in Excel 2016 according to the criteria described in the figure legends. Venn diagrams were created with tools provided at http://bioinformatics.psb.ugent.be/webtools/Venn/. Overrepresentation analyses of gene sets were done using the majority protein IDs (for HuH7 cells) or gene IDs (for Vero E6 cells) of differentially enriched proteins and Metascape software with the express settings[44]. Coregulated proteins in HuH7 and Vero E6 cells were identified based on NCBI gene ID annotation corresponding to the majority protein IDs. Protein network data of filtered gene ID lists were extracted from STRING (version 10, https://string-db.org/)[108] and networks were visualized with Cytoscape 3.8.0[109]. Mapping of ratio values on KEEG pathway hsa04141 was done with Pathview Web software or the Pathview R-package 1.18.2 and R 3.4.4 (https://pathview.uncc.edu/,[110]).

**Statistics, quantification, and reproducibility.** Quantification of data and statistical parameters (means, t-tests, standard variations, confidence intervals, Pearson correlations, linear regressions, non-linear fittings for EC$_{50}$ and CC$_{50}$ values) were calculated using SigmaPlot 11, DESeq2 (version 1.22.1), GraphPad Prism 5.0 or 8.4.3, Perseus (versions 1.6.10.50 (MERS-CoV) or 1.6.14 (SARS-CoV-2)), ImageLab (versions 5.2.1 or 6.0.1), or Microsoft Excel 2016.

Dot plots of cell viability assay results (Figs. 3b, d, 4d and Supplementary Figs. 2c, 5d, e, 10) show technical replicates according to the numbers of independent experiments indicated in the legends.

All statistical tests for pathway enrichment analyses (Figs. 6e–i, 7d and Supplementary Figs. 6–8) were calculated online by Metascape software (https://metascape.org/) using the ontology sources KEGG Pathway, GO Biological Processes, Reactome Gene Sets, Canonical Pathways, CORUM, TRRUST, DisGeNET, PaGenBase, Transcription Factor Targets, WikiPathways, PANTHER Pathway, and COVID and all genes in the genome as the enrichment background. *P*-values were based on the accumulative hypergeometric distribution and *q*-values were calculated using the Benjamini−Hochberg procedure to account for multiple testings. Terms with a *p*-value < 0.01, a minimum count of 3, and an enrichment factor > 1.5 were collected. Kappa scores were used as the similarity metric for hierarchical clustering on the enriched terms, and sub-trees with a similarity of > 0.3 were considered a cluster. The most statistically significant terms within a cluster were chosen to represent the cluster.

**Reporting summary.** Further information on research design is available in the Nature Research Reporting Summary linked to this article.

## Data availability

The mass spectrometry proteomics data in this study shown in Fig. 6a–d have been deposited to the ProteomeXchange Consortium[111] via the PRIDE partner repository[112] with the dataset identifier PXD021222 (https://doi.org/10.6019/PXD021222). The remaining data generated in this study are provided in the Supplementary Information/Source Data files. Source data are provided with this paper.

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

## Acknowledgements

This work was supported by the following grants from the Deutsche Forschungsgemeinschaft (DFG, German Research Foundation): KR1143/9-2 and ZI618/6-2 (KFO309, P3 (to M.K. and J.Z.), Z1 (to T.H.) P2 (to S.H. and S.B.), project 284237345); TRR81/3 (A07 (to M.L.S.), B02 (to M.K.), project 109546710); SFB1213/2 (B03 (to M.K., M.L.S.), project 268555672); SFB1021/2 (A01 (to J.Z.), A05 (to S.B.), C01 (to M.L.S.), C02 (to M.K.), Z02 (to T.H.), Z03 (to M.K., U.L.), C05 (to S.H.), project 197785619); SFB-TR84 (B02 and B09 to S.H.), GRK 2573 (RP4 (to M.L.S), RP5 (to M.K.), project 416910386) and INST 160/708-1 FUGB (to U.L.). The work was also supported by the German Ministry for Education and Research (RAPID, COVINET, and DZIF TTU 01.806, to J. Z.) and the LOEWE program of the state of Hesse (DRUID, B02 to J.Z. and A01 to S.B.). Work in the laboratories of M.K., S.H. and M.L.S. is also supported by the IMPRS program of the Max Planck Society and the Excellence Cluster Cardio-Pulmonary Institute (EXC 2026: Cardio-Pulmonary Institute (CPI), project 390649896) and the DZL/UGMLC/ILH program. Work of J.Z., M.K., S.H. and S.B. is further supported by the Von-Behring-Roentgen-Stiftung and the Pandemics Network Hesse. We thank Andrea Nist and Thorsten Stiewe of the Genomics Core Facility, Philipps University Marburg, Marburg, Germany, and Philipp Mengel, Institute of Medical Microbiology, Justus Liebig University, Giessen, Germany, for RNA sequencing and Christian Drosten for providing SARS-CoV-2 and MERS-CoV.

## Author contributions

M.S.S., C.M., C.M-B., H.W., B.V.A., N.K. and J.M-S. performed experiments, analyzed, and visualized data. U.L. performed mass spectrometry measurements and together with A.W., M.K. and I.B. analyzed proteomics data sets. N.H. analyzed RNA-seq data. T.H. sequenced SARS-CoV-2. M.K. conceived and supervised the study and analyzed and visualized data. M.K. wrote the initial paper draft. J.Z. supervised and analyzed MERS-CoV and SARS-CoV-2 experiments. J.Z., S.B., S.H., M.L.S., (and all other authors) helped to finalize the paper. All authors approved the submitted version of the paper.

## Funding

## Competing interests

The authors declare no competing interests.
