## [Peer Review File · Nature Communications]

Reviewer comments, first round –

Reviewer #1 (Remarks to the Author):

Shaban et al. have investigated a very novel, timely topic focusing on the role of ER and UPR in regulation of Corona Virus Replication. They provided evidence that Corona viruses infection change the activation of PERK arm of UPR while did not have significant effect of IRE and ATF6 arm. The respected team have used Thapsigargin and was able to inhibit viral infection. The respected team has used 3 different cell lines including lung fibroblasts. They have used RNA seq and Mass spectroscopy, and immunoblotting to confirm their results. Overall the project and methods have been very well designed, they used several confirmatory methods to verify their findings and exclude any potential false positive or negative results. I strongly recommend the following consideration to respected authors:

- 1- It is extremely important the respected team show expression of ACE2 receptor in all cells and identify the role of UPR induction in ACE2 expression and localization using membrane fractionation.
- 2- It is very essential the respected team use human primary epithelial cells or HBE cells and use air liquid interface model to confirm their valuable findings in one of the most important cell targets for corona virus infection.
- 3- It is very important the respected team confirm their findings using using GSK-PERK inhibitor and targeting eif phosphorylation using Salubrinal and see what will happen to Corona virus infection in these cells.
- 4- PERK arm of UPR has regulatory effect of autophagy pathway via regulation of autophagy flux. It is very essential the respected team address how changes in PERK activation via coronavirus infection target autophagy flux and Thapsigargin treatment changes these flux.

Reviewer #2 (Remarks to the Author):

In this contribution, Shaban et al find that thapsigargin inhibits coronavirus replication when applied in the nanomolar range, the cytotoxic concentration being considerably higher. The effects of thapsigargin and coronavirus infection seem to cancel each other out, resulting in a rescue of protein synthesis induced by each of both treatments (CoV: 90% reduction; Tha: 90% reduction; combination: 50% reduction as seen by puromycin pulse labeling). This effect is valid in several cell lines and with several coronaviruses. Pathway analysis of proteins differentially expressed under infection (w/wo Tha, analysis by RNAseq + mass spectrometry) identifies a broad range of effects on cellular metabolism. Among those is ER-quality control and ER-associated protein degradation, both involving HERPUD1. Changes in both pathways correlated with an "antiviral state".

The report is a summary of a very well-executed systems biology approach. I find it extremely difficult to make suggestions for improvement of the experiments as conducted. However, I also find it difficult to derive a mechanistic concept that goes beyond ideas like imbalance of major cellular pathways, antiviral state, etc.

It would be helpful to better understand how specific the effect is on viruses other than coronaviruses, and what perspective these findings may offer in terms of therapeutic exploitation. Also, the direct interactions of coronavirus protein products with ER stress components would be of interest to understand the specificity of the observed antiviral effects. Much more testing of cell viability would also help to convince about the effect.

Authors mention in the discussion that Thapsigargin or other substances working on ER stress are under study against cancer. Would it be possible to conduct animal experiments looking at the effect of such substances on viral infection?

Reviewer #3 (Remarks to the Author):

In this manuscript the authors present convincing evidence that chemical induction of ER -stress using thapsigargin suppresses replication of SARS-CoV-2 and the related coronaviruses MERS-CoV and 229E.

I think this is an interesting and useful addition to the field, the one big drawback I suppose is the fact (noted by the authors) that this drug is only in very very early stages of being used in clinical trials for anything and a quick look up suggests that this drug has some serious side effects. Nonetheless, it is certainly an area worth pursuing especially given the low effective dose in cell culture. I think the next obvious step is to use this in an animal model (eg ferrets or hamsters).

I have a couple of observations, large parts of the bioinformatic analysis should be moved to the supplementary datasets as they look nice but in my opinion they are of little practical use in a manuscript figure. The compelling data is in things like the western blots. Secondly, they use VeroE6 cells to grow SARS-CoV-2 which have been shown to select for deletions in and around the furin cleavage site in the spike protein in some cases. Can they check that this is not the case for their virus? I would be surprised if this affected the outcome of their experiments but it is not impossible.

Also I think it would be helpful for the authors to say something about the effects (e.g. dose, fold reduction in virus replication etc) Remdesivir has on the virus in cell culture to put this in context. Remdesivir is potently antiviral but yet does not affect mortality rates so I think this will be useful to put a wider context on the challenge faced by teams searching for antiviral drugs.

REVIEWER COMMENTS

Reviewer #1 (Remarks to the Author):

Shaban et al. have investigated a very novel, timely topic focusing on the role of ER and UPR in regulation of Corona Virus Replication. They provided evidence that Corona viruses infection change the activation of PERK arm of UPR while did not have significant effect of IRE and ATF6 arm. The respected team have used Thapsigargin and was able to inhibit viral infection. The respected team has used 3 different cell lines including lung fibroblasts. They have used RNA seq and Mass spectroscopy, and immunoblotting to confirm their results. Overall the project and methods have been very well designed, they used several confirmatory methods to verify their findings and exclude any potential false positive or negative results. I strongly recommend the following consideration to respected authors:

Reply: We are pleased that this reviewer considers the topic of the study interesting and are grateful for the constructive and helpful suggestions made.

1- It is extremely important the respected team show expression of ACE2 receptor in all cells and identify the role of UPR induction in ACE2 expression and localization using membrane fractionation.

Response: To address this point, we analyzed the ACE2 protein levels in both HuH7 and Vero E6 cells. In line with previous studies, SARS-CoV-2 infection was found to reduce cellular ACE2 protein levels. However, we did not find a negative or positive regulatory effect of thapsigargin on ACE2 protein expression in cells infected with any of the three CoV (see Fig. 1 for review), suggesting that changes in ACE2 levels are unlikely to account for the antiviral effects of thapsigargin. We therefore decided not to include these data in the present manuscript.

2- It is very essential the respected team use human primary epithelial cells or HBE cells and use air liquid interface model to confirm their valuable findings in one of the most important cell targets for corona virus infection.

Response: We appreciate this important suggestion and have established cell culture models of normal human bronchial epithelial (NHBE) cells. These cells were differentiated at air-liquid interphase for several weeks and then infected at three time points in the presence / absence of two doses of thapsigargin. The results confirm (for all three CoVs tested) the potent inhibition of replication in NHBE cells in the presence of thapsigargin (new Fig. 4).

3- It is very important the respected team confirm their findings using using GSK-PERK inhibitor and targeting eif phosphorylation using Salubrinal and see what will happen to Corona virus infection in these cells.

Response: As suggested by the reviewer, we treated cells with 0.1, 1 and 10 μ M of the new PERK inhibitor (PERKi) GSK2656157 using dose ranges that were previously shown to reduce eIF2 α phosphorylation in cultured cells¹. In line with that previous study, we found strong suppression of PERK (auto)phosphorylation as well as of phosphorylation of the PERK substrate eIF2 α at S51 (Fig. S2A, D). At PERKi concentrations between 10 – 50 μ M, we observed a suppression of viral replication as assessed by plaque assay and N protein levels (Fig. S2). These data provide additional independent evidence for a role of the PERK-eIF2 α pathway in CoV replication. Noteworthy, the PERKi did not affect thapsigargin-mediated suppression of viral replication, placing the thapsigargin-mediated antiviral effects downstream of PERK. These results are presented in a new

Fig. S2 and are discussed in the text.

Similarly, we used salubrinal at concentrations of 0.5, 5 and 50 μ M according to previous studies^{2,3}. Salubrinal did not affect cell viability. At the highest dose and longest pre-incubation times tested, we did not find any effects of salubrinal on viral replication (see Fig. 2 for review). We therefore decided not to include these data in the already very large manuscript.

4- PERK arm of UPR has regulatory effect of autophagy pathway via regulation of autophagy flux. It is very essential the respected team address how changes in PERK activation via coronavirus infection target autophagy flux and Thapsigargin treatment changes these flux.

Response: We thank the reviewer for this important comment, which turned to out to add a new facet to our story. Overall, the contribution of (macro)autophagy to CoV replication is controversial with evidence for both pro- and antiviral roles depending on virus strains / model systems used and there were no data for HCoV-229E available at the time of our study^{4,5}. We determined the autophagic flux by measuring the protein amounts of p62 / SQSTM1 and of the lipidated form of LC3, LC3B-II, in the presence or absence of the lysosomal inhibitor bafilomycin A₁ according to the standard protocols used in the field⁶⁻⁸. These data show that HCoV-229E stimulates selective autophagic flux through the formation of p62 / SQSTM1 foci (see Fig. 7). Thapsigargin upregulates p62 / SQSTM1 and LC3B-II levels in HuH7 and Vero E6 cells infected with HCoV-229E, MERS-CoV and SARS-CoV-2 (see Fig. 8). However, all thapsigargin effects on autophagy factors were insensitive to bafilomycin A₁, in line with a previous report showing that thapsigargin initiates early stages of autophagosome formation, but blocks the subsequent maturation and fusion with lysosomes^{9,10}. These results are presented in the new Fig. 7 and in Fig. 8. Also, in a new paragraph added to the discussion on page 12, we now discuss the contribution of the block of autophagic flux to thapsigargin's antiviral effects.

Reviewer #2 (Remarks to the Author):

In this contribution, Shaban et al find that thapsigargin inhibits coronavirus replication when applied in the nanomolar range, the cytotoxic concentration being considerably higher. The effects of thapsigargin and coronavirus infection seem to cancel each other out, resulting in a rescue of protein synthesis induced by each of both treatments (CoV: 90% reduction; Tha: 90% reduction; combination: 50% reduction as seen by puromycin pulse labeling). This effect is valid in several cell lines and with several coronaviruses. Pathway analysis of proteins differentially expressed under infection (w/wo Tha, analysis by RNAseq + mass spectrometry) identifies a broad range of effects on cellular metabolism. Among those is ER-quality control and ER-associated protein degradation, both involving HERPUD1. Changes in both pathways correlated with an "antiviral state".

The report is a summary of a very well-executed systems biology approach. I find it extremely difficult to make suggestions for improvement of the experiments as conducted. However, I also find it difficult to derive a mechanistic concept that goes beyond ideas like imbalance of major cellular pathways, antiviral state, etc.

It would be helpful to better understand how specific the effect is on viruses other than coronaviruses, and what perspective these findings may offer in terms of therapeutic exploitation. Also, the direct interactions of coronavirus protein products with ER stress components would be of interest to understand the specificity of the observed antiviral effects. Much more testing of cell viability would also help to convince about the effect.

Response: We thank this reviewer for her/his interest in our results and for stating that the

experiments and analyses have been well executed. We agree that our data do not reveal a precise mechanism of how thapsigargin mediates its antiviral effects. However, (i) the broad effects of this compound on the proteome (this study) and (ii) the multiple effects on the transcriptomic level observed previously in cells exposed to chemical stress (including thapsigargin) clearly implicated an atypical activation of ER stress / the UPR in this process^{11,12}. Together, these data lead us to suggest that there will be no simple thapsigargin-regulated linear pathway. The full elucidation of the antiviral mechanisms will have to be worked out in future studies.

To address the reviewer's comment on potential virus-specific effects of thapsigargin, we analyzed the effects of thapsigargin on influenza A virus and poliovirus, respectively. As shown in Fig. S4, thapsigargin potently inhibits IAV but not poliovirus replication, suggesting that, in addition to CoVs, this compound may also inhibit other enveloped RNA viruses, in line with a previous study showing that the replication of several paramyxoviruses (e.g., peste des petits ruminants virus and Newcastle disease virus) is suppressed by thapsigargin while replication of a poxvirus included in that study was not affected¹³.

Concerning the possible direct interactions of ER stress components with CoV proteins, we believe that it will be important to perform proximity-based labelling of interactors in CoV-infected cells using BioID or related approaches that will also work in the environment of the ER. These experiments are technically challenging and we feel that they are beyond the scope of the present study. However, we now include in the discussion recent interaction studies of individually expressed CoV proteins which reported some ER-related interaction partners¹⁴⁻¹⁶. At present, the physiological relevance of these findings remains elusive. In the long-term, comprehensive interaction studies need to be performed in virus-infected cells in order to recapitulate (only) interactions that occur in cells in which the full range of viral proteins is expressed and the typical virus-induced membranous replication compartments (DMVs, CM) are being formed.

To address the point of additional cell viability testing, we followed the fate of cells after a single-dose usage of thapsigargin for 24, 48 and 72 h to provide more evidence that the compound exerts long-lasting antiviral effects while improving survival (new Fig. 1J).

We also used an alternative ATP-dependent viability assay (Fig. 3K) and performed several additional MTT/MTS assays that are now shown in Fig. S2C, Fig. S5D-E, Fig. S10 and Fig. 2 for review (see below). We also assessed the viability of thapsigargin-treated NHBE cells by TEER assay (Fig. S5F).

Authors mention in the discussion that Tha derivatives or other substances working on ER stress are under study against cancer. Would it be possible to conduct animal experiments looking at the effect of such substances on viral infection?

Reponse: We agree that it would be the next logical step to translate our findings into a potential clinical application. However, for several reasons, these experiments are impossible for us to do during this revision: (i) world-wide, only few laboratories have established animal models suitable to study HCoV-229E, MERS-CoV or SARS-CoV-2, (ii) our laboratories do not have this expertise and we have no laboratory setup for suitable animal models, (iii) the thapsigargin variant that has been tested in clinical studies, mipsagargin, is a pro-drug that needs to be activated by tissue-specific proteases present in the prostate and in the endothelium¹⁷. The idea behind this approach is to reduce systemic toxicity. In order to test antiviral effects of thapsigargin in mice (or hamsters or ferrets), we believe that different thapsigargin analogs would have to be designed that, ideally, can be activated by enzymes specific for (nasal, bronchial, lung) epithelial cells as these are the primary replication sites for CoVs. In summary, this type of experiment will take many months and is at

present not feasible for our groups. However, we hope that our results will motivate other groups with specialist expertise in CoV animal models to embark on thapsigargin or develop derivatives suitable for in vivo use. We have highlighted the need to validate the thapsigargin effects in pre-clinical studies in the discussion. We hope that the editors and reviewers will understand these restrictions and will accept our arguments.

Reviewer #3 (Remarks to the Author):

In this manuscript the authors present convincing evidence that chemical induction of ER -stress using thapsigargin suppresses replication of SARS-CoV-2 and the related coronaviruses MERS-CoV and 229E.

I think this is an interesting and useful addition to the field, the one big drawback I suppose is the fact (noted by the authors) that this drug is only in very very early stages of being used in clinical trials for anything and a quick look up suggests that this drug has some serious side effects. Nonetheless, it is certainly an area worth pursuing especially given the low effective dose in cell culture. I think the next obvious step is to use this in an animal model (eg ferrets or hamsters).

Response: We thank this reviewer for stating that the evidence provided in our study is convincing and that the implications are interesting. We agree that the next step towards translation into the clinics will be the testing in animals. However, due to the limitations outlined in the response to reviewer #2 we are not able to perform such animal work for the revision in time. However, as outlined in our response to reviewer #1, we now added infection experiments with a relevant ex vivo infection model using primary human NHBE cells (new Fig. 4).

I have a couple of observations, large parts of the bioinformatic analysis should be moved to the supplementary datasets as they look nice but in my opinion they are of little practical use in a manuscript figure. The compelling data is in things like the western blots. Secondly, they use VeroE6 cells to grow SARS-CoV-2 which have been shown to select for deletions in and around the furin cleavage site in the spike protein in some cases. Can they check that this is not the case for their virus? I would be surprised if this affected the outcome of their experiments but it is not impossible.

Response: We have considered moving the bioinformatics data sets, but we felt that this will disrupt the visualization of the analysis work flow that eventually led to the discovery of ERAD / autophagy factors as thapsigargin targets (shown in Fig. 6). However, in the light of this comment we have extended the immunoblot results previously shown in Fig. 4F and have added new data which nicely complement and confirm the findings made by mass spectrometry (new Fig. 8). For readers more interested in the global effects of thapsigargin, this will demonstrate the validity of the proteomics approach and we have thus decided to keep the two proteomics figures 5 and 6 in the main figure section (with the majority of the proteomics / bioinformatics data sets still remaining in the supplement). We re-sequenced the genome RNA of the SARS-CoV-2 virus preparation used in this study and confirmed the absence of the multibasic furin cleavage site. The genome sequence has been deposited under the following link: <https://www.ncbi.nlm.nih.gov/sra/?term=PRJNA6582424>.

Also I think it would be helpful for the authors to say something about the effects (e.g. dose, fold reduction in virus replication etc) Remdesivir has on the virus in cell culture to put this in context. Remdesivir is potently antiviral but yet does not affect mortality rates so I think this will be useful to put a wider context on the challenge faced by teams searching for antiviral drugs.

Response: To address this important request, we have compared thapsigargin and remdesivir side by side and assessed replication (by plaque assay and RT-qPCR) and cell viability (new Fig. S5). The results show that remdesivir has comparable efficacy for HCoV-229E and MERS-CoV in our systems but is significantly less efficient than thapsigargin for SARS-CoV-2.

Data availability for review:

The mass spec data set is available for reviewers here:

<https://www.doi.org/10.6019/PXD021222>

Reviewers log in at <https://www.ebi.ac.uk/pride/login>

username: reviewer08330@ebi.ac.uk

password: HRxwc2LR.

Fig. 1 for review. Expression of ACE2 protein in CoV-infected cells.

- (A) Whole cell extracts from HuH7 cells infected with HCoV-229E or MERS-CoV and from Vero E6 cells infected with SARS-CoV-2 in the presence or absence of thapsigargin (1 μ M) were analyzed for the expression of ACE2 by immunoblotting using the polyclonal goat antibody AF933 from R&D Systems.
- (B) Quantification of ACE2 protein levels from four replicates of MERS-CoV- or SARS-CoV-2-infected cells. SARS-CoV-2 infection reduces total ACE2 protein levels in Vero E6 cells. However, there is no effect of thapsigargin on ACE2 levels in any of the infected cells.

Fig. 2 for review. The selective inhibitor of eIF2 α dephosphorylation salubrin does not suppress CoV replication.

- (A) Scheme of salubrin treatments of HuH7 cells. The dose of salubrin (50 μ M, obtained from Cayman Chemicals; Cay14735) and the forty hour treatment were chosen according to (Boyce et al., Science 2005).
- (B) HCoV-229E viral titers in cells treated with salubrin alone or with thapsigargin added together with the infection or 8 h p.i..
- (C) Huh7 cells were pretreated with 50 μ M salubrin according to the scheme shown in (A), or were treated with solvent (DMSO), or were left untreated. Then, cells were infected with HCoV-229E (MOI = 1) as indicated. Thapsigargin was added simultaneously with the infection or 8 h p.i.. Cell extracts were analyzed by immunoblotting for the phosphorylation or expression of the indicated proteins. The lower graphs show quantification of two independent immunoblots as shown in (C) demonstrating the increase in serine 51 phosphorylation of eIF2 α .
- (D) Viability (by MTS assays) of cells exposed to increasing doses of salubrin for 48 h in the presence / absence of viral infection within the last 24 h.

References:

- 1 Krishnamoorthy, J. *et al.* Evidence for eIF2alpha phosphorylation-independent effects of GSK2656157, a novel catalytic inhibitor of PERK with clinical implications. *Cell Cycle* **13**, 801-806, doi:10.4161/cc.27726 (2014).
- 2 Nakajima, S., Chi, Y., Gao, K., Kono, K. & Yao, J. eIF2alpha-Independent Inhibition of TNF-alpha-Triggered NF-kappaB Activation by Salubrinal. *Biol Pharm Bull* **38**, 1368-1374, doi:10.1248/bpb.b15-00312 (2015).
- 3 Boyce, M. *et al.* A selective inhibitor of eIF2alpha dephosphorylation protects cells from ER stress. *Science* **307**, 935-939, doi:10.1126/science.1101902 (2005).
- 4 Miller, K. *et al.* Coronavirus interactions with the cellular autophagy machinery. *Autophagy*, 1-9, doi:10.1080/15548627.2020.1817280 (2020).
- 5 Fecchi, K. *et al.* Coronavirus Interplay With Lipid Rafts and Autophagy Unveils Promising Therapeutic Targets. *Front Microbiol* **11**, 1821, doi:10.3389/fmicb.2020.01821 (2020).
- 6 Klionsky, D. J. *et al.* Guidelines for the use and interpretation of assays for monitoring autophagy (4th edition). *Autophagy*, 1-382, doi:10.1080/15548627.2020.1797280 (2021).
- 7 Yoshii, S. R. & Mizushima, N. Monitoring and Measuring Autophagy. *Int J Mol Sci* **18**, doi:10.3390/ijms18091865 (2017).
- 8 Jiang, P. & Mizushima, N. LC3- and p62-based biochemical methods for the analysis of autophagy progression in mammalian cells. *Methods* **75**, 13-18, doi:10.1016/j.ymeth.2014.11.021 (2015).
- 9 Ganley, I. G., Wong, P. M., Gammoh, N. & Jiang, X. Distinct autophagosomal-lysosomal fusion mechanism revealed by thapsigargin-induced autophagy arrest. *Mol Cell* **42**, 731-743, doi:10.1016/j.molcel.2011.04.024 (2011).
- 10 Ganley, I. G., Wong, P. M. & Jiang, X. Thapsigargin distinguishes membrane fusion in the late stages of endocytosis and autophagy. *Autophagy* **7**, 1397-1399, doi:10.4161/auto.7.11.17651 (2011).
- 11 Bergmann, T. J. & Molinari, M. Three branches to rule them all? UPR signalling in response to chemically versus misfolded proteins-induced ER stress. *Biol Cell* **110**, 197-204, doi:10.1111/boc.201800029 (2018).
- 12 Bergmann, T. J. *et al.* Chemical stresses fail to mimic the unfolded protein response resulting from luminal load with unfolded polypeptides. *J Biol Chem* **293**, 5600-5612, doi:10.1074/jbc.RA117.001484 (2018).
- 13 Kumar, N. *et al.* Inhibitor of Sarco/Endoplasmic Reticulum Calcium-ATPase Impairs Multiple Steps of Paramyxovirus Replication. *Front Microbiol* **10**, 209, doi:10.3389/fmicb.2019.00209 (2019).
- 14 St-Germain, J. R. *et al.* A SARS-CoV-2 BioID-based virus-host membrane protein interactome and virus peptide compendium: new proteomics resources for COVID-19 research. *bioRxiv*, doi:10.1101/2020.08.28.269175 (2020).
- 15 Laurent, E. M. N. *et al.* Global BioID-based SARS-CoV-2 proteins proximal interactome unveils novel ties between viral polypeptides and host factors involved in multiple COVID19-associated mechanisms. *bioRxiv*, doi:10.1101/2020.08.28.272955 (2020).
- 16 Samavarchi-Tehrani, P. *et al.* A SARS-CoV-2 - host proximity interactome. *bioRxiv*, doi:10.1101/2020.09.03.282103 (2020).
- 17 Mahalingam, D. *et al.* A Phase II, Multicenter, Single-Arm Study of Mipsagargin (G-202) as a Second-Line Therapy Following Sorafenib for Adult Patients with Progressive Advanced Hepatocellular Carcinoma. *Cancers (Basel)* **11**, doi:10.3390/cancers11060833 (2019).

Reviewer comments, second round –

Reviewer #1 (Remarks to the Author):

I like to congratulate all respected team to be able to do a fantastic revision. The revised manuscript shows the impact of the cross talk of ER-Stress/UPR with autophagy in regulation for cellular response to SARS-CoV-2 infection. This will certainly brings outstanding opportunity to understand the mechanism of infection and potential new therapeutic strategies for SARS family infection targeting UPR/autophagy. I do not have any further comments.

Saeid Ghavami, PhD

Associate Professor of Cancer Biology, Department of Human Anatomy and Cell Science, University of Manitoba

Reviewer #2 (Remarks to the Author):

Authors have followed my recommendation to include other viruses so the reader obtains an impression about how specific the rather complex mechanism is. Their explanations for not including further suggested experiments (animal studies, virus protein-specific studies) are convincing. I have no further comments.

Reviewer #1 (Remarks to the Author):

I like to congratulate all respected team to be able to do a fantastic revision. The revised manuscript shows the impact of the cross talk of ER-Stress/UPR with autophagy in regulation for cellular response to SARS-CoV-2 infection. This will certainly brings outstanding opportunity to understand the mechanism of infection and potential new therapeutic strategies for SARS family infection targeting UPR/autophagy. I do not have any further comments.

Saeid Ghavami, PhD

Associate Professor of Cancer Biology, Department of Human Anatomy and Cell Science, University of Manitoba

Reply: We are pleased that Dr. Ghavami is satisfied with our revised manuscript. We would like to take this opportunity to thank him for his fair review, the important suggestions that helped improve our work, and for disclosing his identity.

Reviewer #2 (Remarks to the Author):

Authors have followed my recommendation to include other viruses so the reader obtains an impression about how specific the rather complex mechanism is. Their explanations for not including further suggested experiments (animal studies, virus protein-specific studies) are convincing. I have no further comments.

Reply: We thank the reviewer for acknowledging our revised work and for the important comments and suggestions in the first round of review.